# Intelligent single-shot full-field characterization over femtosecond pulses

Guoqing Pu ⊠, Chao Luo, Weisheng Hu & Lilin Yi ⊠

Conventional approaches to femtosecond (fs) pulse characterization depend on nonlinearities without exception, thereby impeding their utilization in measuring weak fs pulses. Their reliance on iterative phase retrieval or spectrometer-based readout further prevents single-shot full-field characterization over high-repetition-rate fs pulse trains. Here, we present linear spectral shearing interferometry (LSSI) based intelligent single-shot full-field characterization (ISFC) towards high-repetition-rate fs pulse trains, where both intensity and phase of fs pulses are reconstructed in single shot from temporal interferograms through a well-trained fully-connected neural network. The spectral shear in LSSI is created completely by linear effects enabling measurement of weak fs pulses. We validate the proposed method on a megahertz fs pulse train with picojoule-level energy. Moreover, the switching dynamics of a programmable spectral filter are successfully resolved with the proposed method at a megahertz frame rate. We anticipate LSSI-based ISFC to be a particularly powerful tool for characterizing weak ultraviolet fs pulses, and even attosecond pulses with high repetition rates.

Accurate characterization over femtosecond (fs) pulses is vital to various utilizations of fs lasers, incorporating ultrafast spectroscopy[1], high-precision ranging[2,3], nonlinear optics[4–6], and attosecond science[7,8]. A subtle idea for fs pulses characterization is to split the pulse under characterization into two arms sent to a nonlinear crystal for optical autocorrelation[9], which can only shed approximate temporal information of the pulse. Frequency-resolved optical gating (FROG) is then proposed to spectrally resolve the optical auto-correlation signal, and a two-dimensional spectrogram (i.e., the FROG trace) is recorded[10]. The iterative-Fourier-transform algorithm can be applied for retrieving both the intensity and phase (i.e., full-field characterization) of the pulse from the measured FROG trace[11]. Time lens and dispersive Fourier transform (DFT) are utilized together for full-field characterization over picosecond pulses by performing the iterative Gerchberg-Saxton algorithm between temporally magnified pulse profiles and real-time spectra[12,13]. Another prominent approach to non-iterative full-field characterization over fs pulses and even attosecond pulses[14] is spectral phase interferometry for direct electric-field reconstruction (SPIDER), where the spectral shear is created by

sum-frequency generation in a very sophisticated setup[15,16]. In recent years, phase-preserving nonlinear autocorrelation[17] and nonlinear-ultrafast-temporal gate-based direct sampling[18,19] have been demonstrated to deliver full-field characterization over fs pulses. However, the above methods all rely on nonlinearities, thus the demand on the pulse energy is non-negligible, especially when it comes to single-shot measurement (e.g., the microjoule level for FROG[16,20]). On the other hand, most of the above methods require iterative algorithms for phase retrieval, preventing single-shot full-field characterization over fs pulse trains with high repetition rates. SPIDER delivers non-iterative phase retrieval, but the spectral interferogram readout through the spectrometer imposes restrictions on its measurement speed.

Linear spectral shearing interferometry (LSSI) is utilized for characterization over ultrashort pulses and also delivers direct phase retrieval[21–23]. The spectral shear in LSSI is introduced by mapping the temporal delay to a spectral frequency difference through DFT, and the measurable pulse energy can be substantially reduced due to the lack of nonlinearities. An implicit but imperative advantage of LSSI is that the interferogram generated in spectral shearing interferometry

State Key Laboratory of Photonics and Communications, School of Information Science and Electronic Engineering, Shanghai Jiao Tong University, Shanghai, China. ⊠e-mail: teddyghf1994@sjtu.edu.cn; lilinyi@sjtu.edu.cn

of a pulse train is converted to consecutive temporal interferograms, thereby allowing single-shot characterization. Pulse reconstruction through the Fourier-transform-based method (FTM)[21–23] in LSSI extremely depends on precise calibration over the time delay of the interferometer and dispersion parameters of DFT, which fluctuate subjected to environmental disturbances[24]. The nonlinear spectral-to-temporal mapping induced by inevitable high-order dispersions can further deteriorate the performance of the FTM when reconstructing pulses with shorter durations[22]. Therefore, accurate single-shot full-field characterization over fs pulses using LSSI in an open environment remains a tough challenge. Until now, there is no single-shot full-field characterization over fs pulse trains with high repetition rates and low energy, therefore limiting the widespread applications of fs pulse characterization.

Here, we combine LSSI and artificial intelligence to provide an effective solution, LSSI-based intelligent single-shot full-field characterization (ISFC), to achieve single-shot full-field characterization over fs pulse train with the high repetition rate and low energy, where fs pulses are reconstructed in single shot from the acquired temporal interferograms in LSSI through a fully-connected neural network. Data-driven ISFC can learn the delay and dispersion values and even their fluctuations through massive data, thereby waiving the laborious calibration of these parameters. Numerical simulations indicate high-order dispersions evoked by nonlinear spectral-to-temporal mapping barely impair pulse reconstruction of ISFC, and ISFC allows to use smaller shear values than the FTM, thereby lowering requirements on the sequential temporal acquisition device (e.g., the bandwidth, the sampling rate).

We experimentally validate LSSI-based ISFC for characterizing a homemade programmable fs pulse laser source. It turns out LSSI-based ISFC realizes single-shot full-field characterization over various fs pulses at the average pulse energy of merely ~296.6 pJ. Moreover, ISFC provides decent pulse reconstruction under low sampling rates.

Given the sampling rate of 3.13 GSa/s, the switching dynamics of the programmable spectral filter are captured and resolved by ISFC in a single-shot full-field manner at the frame rate of ~32.4 MHz. Note that nonlinearities generation in the ultraviolet band is rather difficult due to the strong absorption of common nonlinear crystals in the ultraviolet band[25–27]. Hence, LSSI-based ISFC, entirely relying on linear effects, is very promising in characterizing weak ultraviolet fs pulses and even attosecond pulses. On the other hand, LSSI-based ISFC could serve as a powerful tool for capturing fast, non-repeatable events at high frame rates.

## Results

### ISFC over fs pulses via LSSI

Figure 1 conceptually illustrates the LSSI-based single-shot full-field characterization over fs pulses via conventional FTM and ISFC. The pulse under characterization is first sent to an interferometer, where spectral interference of the pulse under characterization and its time-delayed replica occurs. Then, the time delay is converted to a spectral shear through the temporal-to-spectral mapping built on DFT. The LSSI consists of the interferometer and DFT. Unlike the spectral shear created by nonlinearities in SPIDER, the spectral shear here is created by two linear effects only, interferometry and dispersion. Since the LSSI only relies on linear effects, the order of the two elements can be swapped. When the spectral shear is introduced, the other subtle but imperative effect of DFT is that the spectral fringes are also mapped as temporal interferograms. As a result, consecutive temporal inter-ferograms can be acquired by a high-speed real-time oscilloscope after photodetection, thereby enabling sing-shot pulse characterization. In LSSI, the spectral shear created by the time delay and DFT can be expressed as

$$\Delta\omega = \frac{\Delta t}{\beta_2 L}. \tag{1}$$

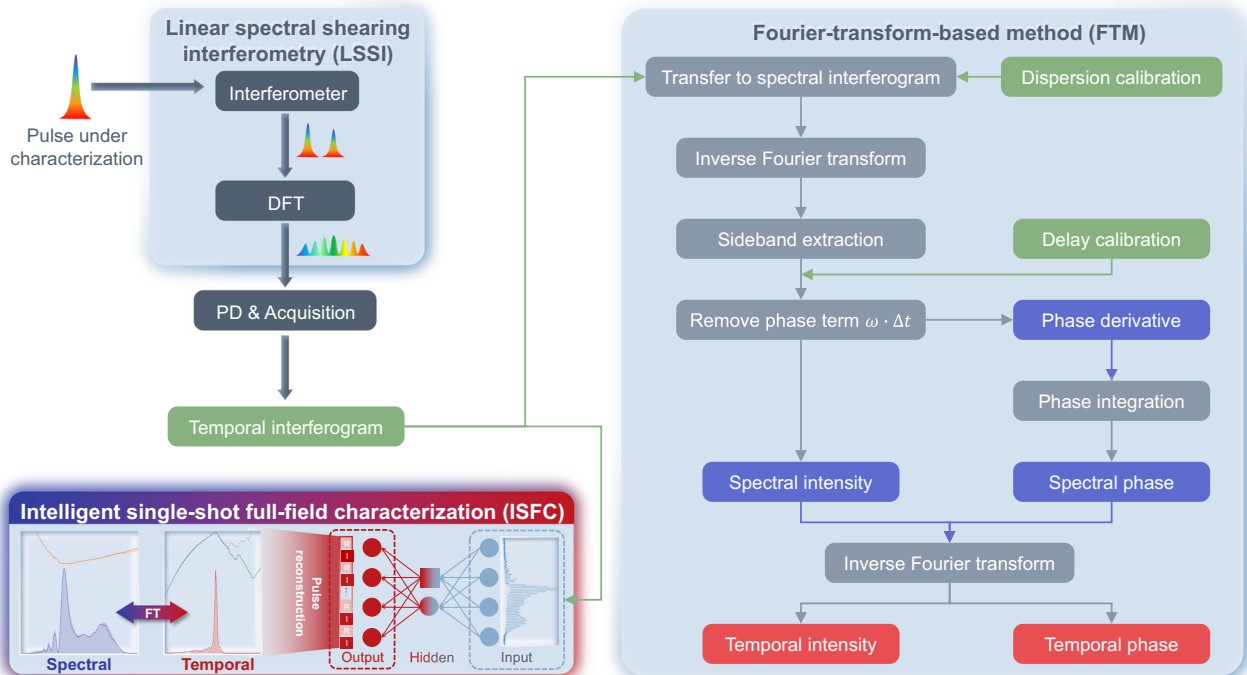

**Fig. 1 | Conceptual illustration of linear spectral shearing interferometry (LSSI) based single-shot full-field characterization over fs pulses via Fourier-transform-based method (FTM) and intelligent single-shot full-field characterization (ISFC).** The pulse under characterization first enters an inter-ferometer, generating spectral interference between the original and its time- delayed replica. Through dispersive Fourier transform (DFT), the delay is converted into spectral shear, and spectral fringes are simultaneously mapped into temporal interferograms. Temporal interferograms are captured via a high-speed oscillo-scope after photodetection, enabling single-shot pulse reconstruction with non-iterative FTM and ISFC.

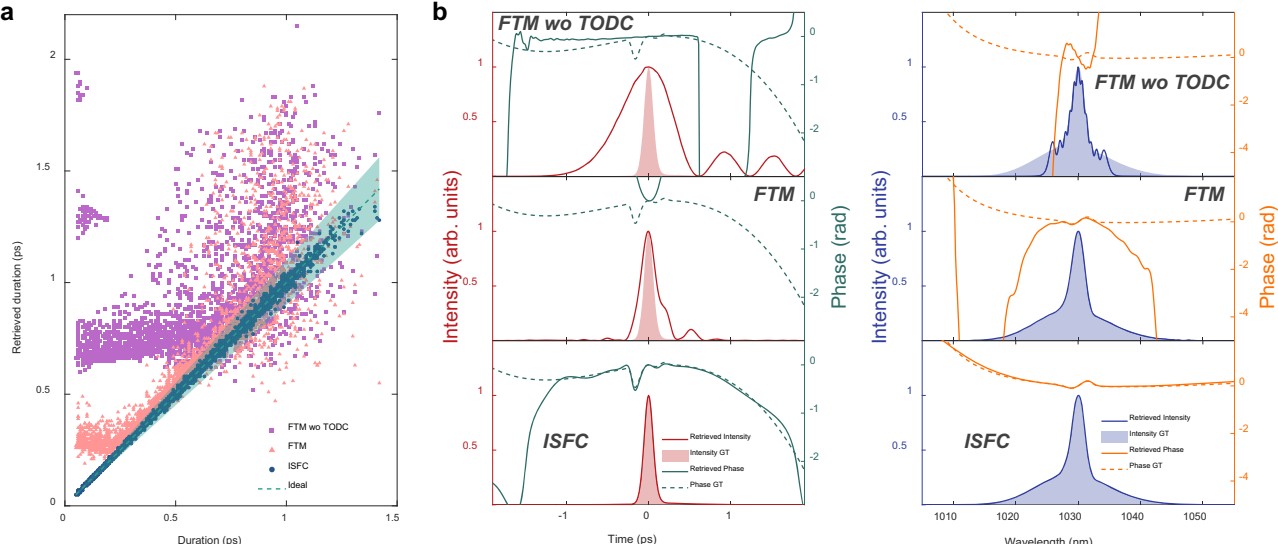

**Fig. 2 | Comparison of FTM and ISFC for fs pulse reconstruction in numerical simulations with the third-order dispersion (TOD). a** Pulse duration retrieval results of FTM without TOD compensation (FTM wo TODC), FTM, and ISFC over the dataset with TOD, where the dashed line and shadow area respectively denote the ideal situation and the 10% error range. **b** Temporal retrieval results (the left column) of three different algorithms over a 130 fs pulse from the test set, and corresponding spectral retrieval results (the right column) via performing Fourier transform on the retrieved temporal pulses.

$\Delta t$ is the delay, $\beta_2$ and $L$ are the group velocity dispersion and length of the dispersion medium, respectively. The interferogram after photo-detection is derived as

$$I_{int}(t) = A(\omega)^2 + A(\omega + \Delta\omega)^2 + A(\omega)A(\omega + \Delta\omega) \\ \left[ e^{j(\omega \cdot \Delta t - \Delta\varphi(\omega))} + e^{-j(\omega \cdot \Delta t - \Delta\varphi(\omega))} \right], \quad (2)$$

where $A(\omega)$ and $A(\omega + \Delta\omega)$ are the spectral amplitude of the original pulse and the shear pulse, respectively. $\Delta\varphi(\omega)$ is the spectral phase difference between the original pulse and the shear pulse. The spectral shear is very small compared to the spectral bandwidth of the pulse under characterization. Hence, $A(\omega + \Delta\omega)$ approximately equals to $A(\omega)$ and the spectral phase can be retrieved by

$$\varphi(\omega) = \frac{\int \Delta\varphi(\omega)d\omega}{\Delta\omega}. \quad (3)$$

Therefore, each sideband of $I_{int}(t)$ centered at the shear includes the desired spectral amplitude and phase information. Conventional FTM first transforms the temporal interferogram back to the spectral domain via the temporal-to-spectral mapping built on DFT, where accurate calibration over dispersion values of DFT is required. Then, one sideband centered at $\Delta t$ or $-\Delta t$ is fetched via inverse Fourier transform, and the linear phase term $\omega \cdot \Delta t$ is removed by using the calibrated delay of the interferometer. Thus, the spectral amplitude $A(\omega)$ and phase derivative $\Delta\varphi(\omega)$ can be obtained. Spectral phase $\varphi(\omega)$ can be calculated via integration on $\Delta\varphi(\omega)$ and thus the spectral field of the pulse under characterization is reconstructed. The corresponding temporal field can be obtained by performing inverse Fourier transform on the reconstructed spectral field.

However, due to the wide spectrum of the fs pulse, high-order dispersions, especially the third-order dispersion (TOD), induced nonlinear temporal-to-spectral mapping is non-negligible. Hence, precise TOD measurement of the dispersive medium is vital to regulate the nonlinear temporal-to-spectral mapping when using the FTM to reconstruct fs pulses. On the other hand, owing to the integration operation in phase retrieval, the FTM is extremely sensitive to shear calibration. The bias in shear calibration leads to a linear slope in the spectral phase derivative $\Delta\varphi(\omega)$ and the linear slope is further

converted to a quadratic phase term (i.e., linear chirp) in the retrieved spectral phase $\varphi(\omega)$, thereby distorting the temporal pulse shape. Precise shear calibration requires accurate measurement of delay and group velocity dispersion. Tracking delay in real time is proposed to reduce the error in shear calibration[22]. Nevertheless, the shear calibration error still exists since the group velocity dispersion fluctuates in an open environment[24]. Therefore, accurate full-field characterization over fs pulses using LSSI via the FTM in an open environment remains a tough challenge.

To circumvent the drawbacks of the FTM, we propose ISFC, where a fully-connected neural network (see Methods for details) reconstructs full-field information of the pulse under characterization from the temporal interferogram in single shot. The delay, dispersion values, and even fluctuations of these parameters of an LSSI system are included in massive data, which can be learnt by ISFC. Hence, data-driven ISFC omits the laborious calibration of the delay and dispersion values. ISFC even manifests decent robustness against fluctuations of key parameters. ISFC is designed to output the interleaved real and imaginary parts of the temporal pulse, and the temporal field can be readily reconstructed. We found that the interleaved sequencing outperformed concatenating the intensity and phase. Through the Fourier transform, the spectral field of the pulse can also be reconstructed. It is equivalent for ISFC to predict the spectral field first and then obtain the temporal field via the Fourier transform.

## Numerical simulations of ISFC over fs pulses via LSSI

We construct a numerical model to emulate the LSSI system and generate simulation datasets for training ISFC (see Methods for details). Since LSSI only relies on linear effects, nonlinearities are not considered in numerical simulations for efficiency improvement. Figure 2a shows the pulse duration retrieval accuracy in the simulation dataset, where TOD in DFT is considered. The FTM without TOD compensation (i.e., FTM wo TODC), where TOD is not considered in spectral interferogram reconstruction, performs worst in the duration retrieval accuracy as expected. Because the nonlinear spectral-to-temporal mapping effect induced by TOD is more severe on shorter pulses with wider spectra, it is obvious that the shorter pulses suffer larger errors. After compensating for the nonlinear mapping induced by TOD in spectral interferogram reconstruction, the situation is well

alleviated as shown in the duration retrieval results of the FTM. When the pulse becomes even shorter, the conventional TOD compensating method cannot completely compensate for the TOD-induced nonlinear mapping effect. As shown in Equation (S1) in Supplementary Information, the high-order dispersions induced nonlinear spectral-to-temporal mapping can be expressed as a polynomial combination between the temporal and spectral coordinates, with the coefficients determined by the dispersion parameters. Thus, the nonlinear polynomial relation can be accurately learnt by ISFC through training on massive data, waiving precise calibration on the high-order dispersion parameters. As a result, ISFC performs perfectly in the entire pulse duration range. Almost all the retrieved durations of ISFC lie in the 10% error range (i.e., indicated by the shadow area). The mean absolute errors (MAEs) of the retrieved durations on the test set of FTM without TOD compensation, FTM, and ISFC are, respectively, 336.83 fs, 127.23 fs, and 8.85 fs. The proposed ISFC is ~14 times better than the conventional FTM in terms of duration retrieval accuracy. Figure 2b demonstrates the reconstructed results of a 130 fs pulse by the three algorithms. In the temporal pulse retrieved by FTM without TOD compensation, the tail oscillations induced by TOD are obvious. With TOD compensation, the tail oscillations are substantially mitigated, and the pulse duration is much closer to the actual duration, but both the temporal intensity and retrieved phase are still far from the ground truth. The reconstructed pulse via ISFC is consistent with the ground truth in both temporal and spectral domains, thereby validating the salient performance of ISFC in compensating the nonlinear spectral-to-temporal mapping effect induced by high-order dispersions.

To evaluate the performance on retrieving the temporal envelope, the duration normalized mean absolute error (NMAE) is introduced and defined as

$$\Gamma_{err} = \frac{1}{N} \cdot \sum_{n=1}^{N} \frac{|\tau_n - \widehat{\tau}_n|}{\widehat{\tau}_n}, \qquad (4)$$

where $\tau_n$ and $\widehat{\tau}_n$ are the retrieved and actual pulse duration for the $n$-th sample in the test set, respectively, and $N$ is the size of the test set. Normalized root-mean-squared error (NRMSE) is used to quantitatively assess the temporal phase retrieval ability, and it is defined as

$$\Phi_{err} = \frac{1}{N} \cdot \sum_{n=1}^{N} \sum_{k=1}^{K} \sqrt{\frac{(\phi_n(k) - \widehat{\phi_n(k)})^2}{K \cdot \widehat{\phi_n^{eff}}^2}}, \qquad (5)$$

where $\phi_n(k)$ and $\widehat{\phi_n(k)}$ denotes the $k$-th point in the retrieved and actual temporal phase curve of the $n$-th sample in the test set, respectively, $K$ is the number of points in the phase curve. $\phi_n^{eff}$ represents the effective phase range of the $n$-th actual temporal phase curve in the test set, which is the absolute phase range in the effective area. The effective area for calculating $\phi_n^{eff}$ is defined as the 20-dB range below the temporal intensity peak. Because there is little energy outside the defined effective area, the corresponding phase is considered trivial to the pulse. Due to thermal and mechanical disturbances, fluctuations of key parameters in an LSSI system are inevitable in the real scene. To evaluate the robustness against random fluctuations of key parameters in LSSI, several simulation datasets with different levels of shear jitter are generated, where high-order dispersions in DFT are ignored. As shown in Fig. 3a, the duration NMAE of ISFC grows more slowly than that of FTM as the shear jitter becomes stronger, indicating ISFC is more robust against shear jitters than FTM.

On the other hand, the shear value is crucial to pulse reconstruction and the subsequent acquisition device. Since the temporal fringe frequency is the shear when high-order dispersions are neglected, the temporal interferogram with the smaller shear can be sampled with a lower-bandwidth and lower-sampling-rate analog-to-digital converter (ADC). Given a fixed dispersive medium, the small shear value corresponds to the small delay, which may be not enough to separate the pulse under measurement and its delayed replica in the time domain. Therefore, the choice of the shear value should consider both the bandwidth and sampling rate of the acquisition system and the approximate pulse duration of the pulse under measurement. The optimal shear should be the minimal one that allows accurate pulse reconstruction. We evaluate the performance of both pulse reconstruction algorithms in various shear values via generating simulation datasets with different delays, where the group velocity dispersion remains fixed and high-order dispersions are still ignored. As shown in Fig. 3b, ISFC distinctly outperforms FTM in all the tested delays, and it signifies that ISFC allows to use smaller shear, thereby lowering the demand on the bandwidth and sampling rate of the acquisition device. Figure 3d shows the retrieved pulse duration results of FTM under different delays. When the delay is 5 ps, small pulse durations are well retrieved since high-order dispersions are not considered. However, as shown in Fig. 3c, the delay is not large enough to avoid overlapping between sidebands and DC components after performing inverse Fourier transform on reconstructed interferograms for wider pulses, thereby resulting in terrible sideband extraction in FTM. As a result, accurate retrieval of pulse durations above ~500 fs is not guaranteed as shown at the top of Fig. 3d. This phenomenon becomes more severe when decreasing the delay to 4 ps. Further reducing the delay to 3 ps even deteriorates the performance of pulses with small durations. In contrast, ISFC accurately retrieves the pulse durations under the three different delays as shown in Fig. 3e. Additional simulations reveal that ISFC achieves significantly better pulse reconstruction performance than FTM when using the lower sampling rate and the worse quantization resolution for temporal interferogram acquisition (see Supplementary Information). Furthermore, ISFC reveals stronger resistance against random noises than FTM in simulations (see Supplementary Information).

## Experimental validation of ISFC over fs pulses via LSSI

Figure 4 shows the experimental setup for validating ISFC over fs pulses via LSSI. A 32.4 MHz programmable fs laser is built using a homemade C-band fs laser followed by a programmable spectral filter. One branch of the shaped pulses is measured by a commercial FROG for label collection, and the other is sent to the LSSI-based ISFC. First, a fiber Mach-Zehnder interferometer is used to obtain a time-delayed replica of the pulse under characterization. Then, the pulse and its time-delayed replica are combined and propagate through a spool of dispersion compensation fiber (DCF) to transfer the delay to a spectral shear. The fiber system before the DCF is polarization-maintaining for better interference. A photodetector converts the temporal interferograms to the electrical domain and is then acquired by an oscilloscope. To apply FTM in the experiments for pulse reconstruction, the shear frequency in experiments is calculated to be ~2.13 GHz by calibrating the delay of the interferometer and dispersion values of the DCF (see Methods and Supplementary Information).

Figure 5a shows the pulse duration retrieval accuracy comparison over the test set. The superiority of ISFC over FTM is further magnified in experiments, manifesting decent pulse duration retrieval from 146 fs to 3.21 ps. The duration MAE on the experimental test set of ISFC is 41.32 fs, which is even smaller than the pre-set temporal resolution of ~70 fs of the FROG in experiments. However, the duration MAE on the experimental test set of FTM reaches 8.16 ps. As shown in Fig. 5c, the corresponding duration NMAE of ISFC and FTM are ~0.07 and 17.12 respectively. In terms of the phase retrieval accuracy, the phase NRMSE of ISFC is 0.0256, which is ~17 folds better than that of FTM. The most responsible reason for the terrible performance of FTM is its inherent sensitivity to the shear calibration, which has been unveiled in previous simulations. Experiments are carried out in an open environment. Thus, during the long time of experimental data acquisition

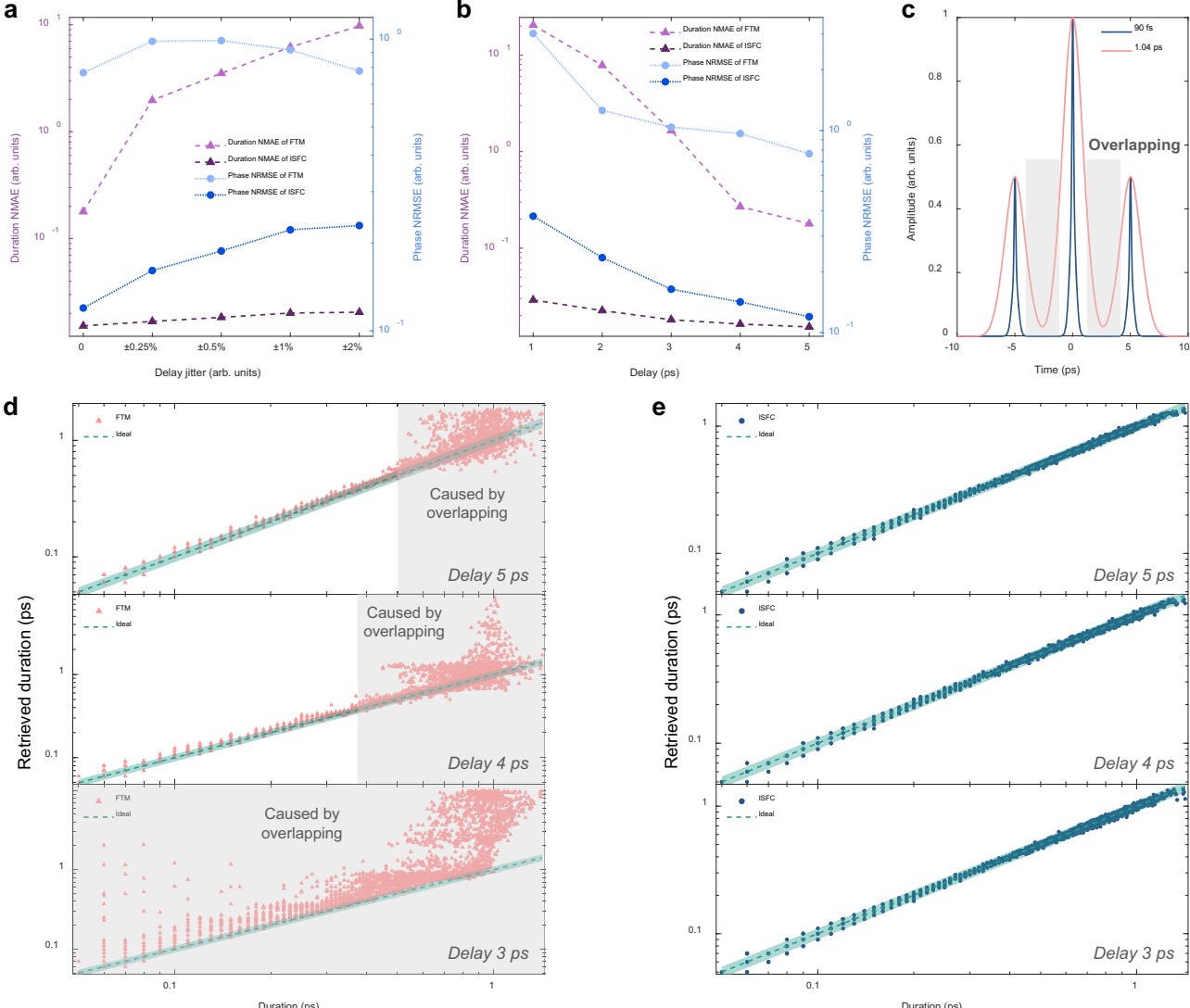

**Fig. 3 | Dependences on delay jitter and delay of ISFC and FTM in numerical simulations. a** ISFC manifests better robustness against delay jitter than FTM. **b** ISFC outperforms FTM under different delays, and ISFC allows to use a smaller delay corresponding to a smaller shear. The smaller delay causes overlapping hindering lossless sideband extraction after inverse Fourier transform in FTM (**c**),

resulting in terrible pulse reconstruction for pulses with larger durations (**d**), but ISFC performs well under smaller delays (**e**). The dashed lines and shadow areas in **d,e** respectively denote the ideal situation and the 10% error range. NMAE, normalized mean absolute error. NRMSE, normalized root-mean-squared error.

time due to the slow updating rate of the FROG (i.e., updating every ~7 s), the delay and dispersion parameters in LSSI determining the shear might vary along the environmental disturbances (e.g., thermal fluctuations and mechanical vibrations). However, the shear value used in FTM for reconstructing diverse pulses is unchanged, leading to awful pulse reconstruction. Another reason behind the terrible performance of FTM is that the shear frequency is too small. As validated in the simulation results shown in Fig. 3, the small shear prevents effective sideband extraction, thus leading to terrible pulse reconstruction of FTM. Additionally, non-negligible nonlinearities in fiber links of LSSI also contribute to the awful pulse reconstruction, as the analysis in the Discussion.

ISFC bypasses the shear calibration by learning the temporal interferogram and its corresponding pulse directly. Shear variations are encoded in different consecutive temporal interferograms and learnt and further adaptively compensated by ISFC. Figure 5b demonstrates accurate pulse reconstruction of ISFC generalized to various pulses. The top row of Fig. 5b shows the accurate reconstruction of a narrow pulse with a duration of 146 fs, and ISFC also

generalizes to pulses with negative chirp or positive chirp as shown in the two middle rows of Fig. 5b. Moreover, ISFC can even reconstruct pulses with sophisticated shapes as shown in the bottom row of Fig. 5b, where oscillations in the temporal tail are perfectly retrieved. In the region of energy distribution, all the reconstructed temporal phase aligns well with FROG measurements. The spectra shown in the right column of Fig. 5b are obtained by performing the Fourier transform on the corresponding temporal predictions. The spectral intensities and phase reconstructed by ISFC are consistent with FROG measurements.

Powerful single-shot pulse reconstruction of ISFC enables resolving the switching dynamics of the programmable spectral filter with a frame rate as high as the repetition rate of the pulse train. The oscilloscope records about 5 million consecutive frames of interferograms with the sampling rate of 3.13 GSa/s, and each frame is fed to ISFC for single-shot pulse reconstruction. To accommodate the low sampling rate, ISFC is retrained using a dataset with the sampling rate of 3.13 GSa/s, which is obtained via down-sampling the original-100-GSa/s temporal interferograms in the digital domain. Figure 6a shows a switching process of the programmable spectral filter resolved by ISFC

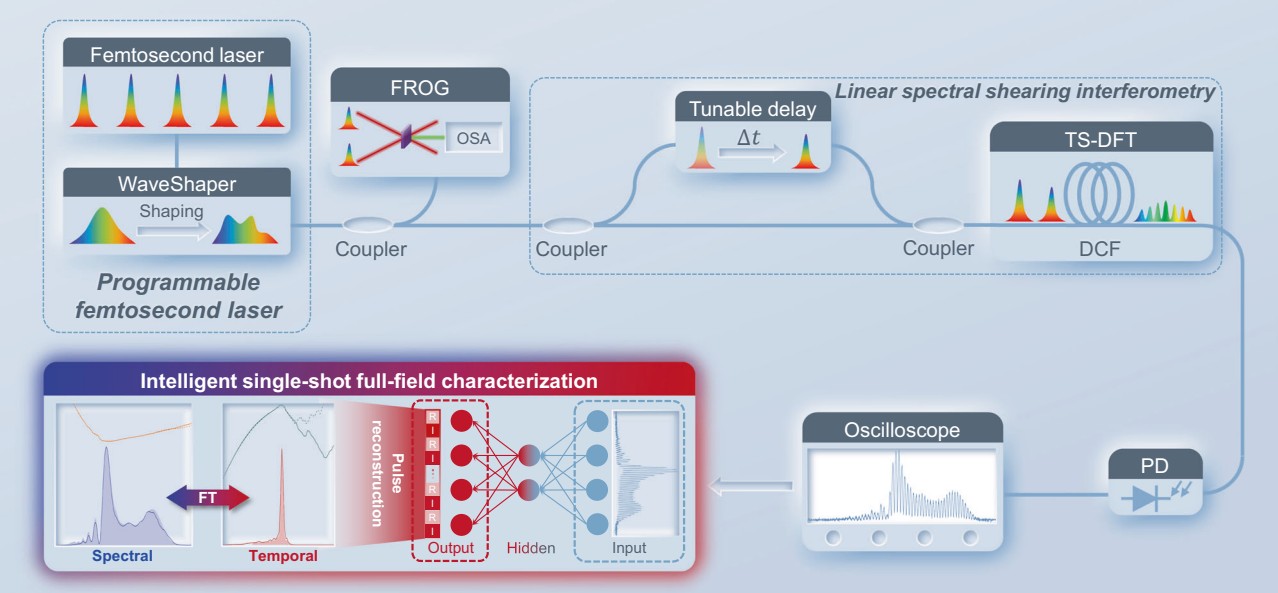

**Fig. 4 | Experimental setup.** The programmable fs laser, composed of a C-band fs laser and a programmable spectral filter, is split into two branches by a fiber coupler. One branch is sent to a commercial frequency-resolved optical gating (FROG) for label collection. The other is sent to LSSI, consisting of a fiber Mach-Zehnder interferometer and a spool of dispersion compensation fiber (DCF) for DFT. A photodetector (PD) converts temporal interferograms generated in LSSI to the electrical domain. An oscilloscope captures temporal interferograms, which are then fed to ISFC for single-shot full-field pulse reconstruction.

in single shot. The left subplot illustrates the intensity dynamics during switching, which starts from the frame of ~1.2 million and terminates at the frame of ~2.7 million. The entire switching process completes within 46.3 ms, conforming to the claimed response time of the programmable spectral filter (less than 100 ms). The pulse first broadens as the switching process initiates, then the peak position of the pulse continues shifting and settles on the main-lobe position of the final pulse. The side lobe of the final pulse originates from the pulse narrowing dynamics observed at the frame of ~2 million. The pulse duration variation during switching can also be recognized in the phase evolution, as shown in the right panel of Fig. 6a. The change in the width of the range, where the phase is meaningful, is synchronized with the change in the pulse duration. The phase of the side lobe and the main lobe of the final pulse almost remains steady after the side lobe settles down. Figure 6b shows the reconstructed pulses of the start frame, the 3 middle frames, and the final frame. Unfortunately, since there is no way to realize full-field characterization over fs pulse train at MHz frame rate, the ISFC resolved full-field switching dynamics cannot be cross validated. However, the validity of the resolved switching dynamics can be guaranteed by for the following two reasons. First, the reconstructed start and final pulses align well with the FROG measurements in both temporal and spectral domains as shown in Fig. 6b. Second, the evolutionary process is consecutive as evidenced in the temporal intensity evolution. Another reconstructed switching dynamics of the programmable spectral filter is demonstrated in Supplementary Information. ISFC provides a viable solution to capture ultrafast events in a single-shot and full-field manner at a frame rate up to MHz. As a result, the LSSI-based ISFC can find utilizations in diverse scenes, such as ultrafast spectroscopy[1] and full-field characterization of ultrafast laser dynamics[12,13,28–30].

By down-sampling experimental temporal interferograms in the digital domain, the sampling rate dependence of ISFC is evaluated and shown in Fig. 7a. Both duration NMAE and phase NRMSE increase as the sampling rate decreases, as expected. Nevertheless, the pulse retrieval performance is acceptable under the sampling rate of 3.13 GSa/s with the duration NMAE of 0.097 and the phase NRMSE of 0.034. The pulse retrieval performance sharply deteriorates when further reducing the sampling rate to 1.56 GSa/s, as evidenced by the

duration NMAE drastically increasing by 104.1% compared to the sampling rate of 100 GSa/s. An empirical deduction drawn from the simulation and experimental results indicates that the minimal sampling rate supported by ISFC lies between the actual shear value and its corresponding Nyquist sampling rate (see Supplementary Information).

The inference time of ISFC for reconstructing one pulse is crucial to implementing online real-time single-shot full-field characterization over high-repetition-rate pulse trains. Figure 7b shows the evaluated inference time for per-pulse characterization using ISFC on both a CPU (Intel Xeon Gold 6146) and a GPU (Nvidia RTX3090) accelerated by compute unified devices architecture (CUDA) under different batch sizes. For each batch size, 1000 tests are executed, and the average inference time is illustrated in Fig. 7b. As batch size increases, the average inference time for per pulse characterization decreases due to parallel computing supported by ISFC. The advantage of the GPU in parallel computing over the CPU is magnified as the batch size increases. The average inference time for per-pulse characterization on the GPU gradually converges to 2.41 us and it represents the supported highest repetition rate of allowing online single-shot full-field characterization via ISFC theoretically reaches ~414 kHz, which could be further improved to MHz level through a more powerful GPU or even ASIC. As a comparison, the FROG in experiments refreshes the pulse characterization result every ~7 s with the FROG trace size of $256 \times 256$, which is astonishingly 3-million-fold slower than ISFC.

## Discussion

We believe the performance gap of ISFC in pulse reconstruction between numerical simulations and experiments is mainly caused by nonlinearities in experiments. The experimental average power before the Mach-Zehnder interferometer is 9.61 mW, and the corresponding average pulse energy is ~296.6 pJ. The shortest pulse in the experimental dataset has a pulse duration of 146 fs, which corresponds to the shortest nonlinear length of merely ~0.37 m. The shortest nonlinear length is far smaller than the length of the following fiber links of the interferometer and DCF. Therefore, nonlinearities are considerable in the LSSI. The large power of the LSSI branch in experiments is chosen for obtaining high-contrast temporal fringes. The average power of the

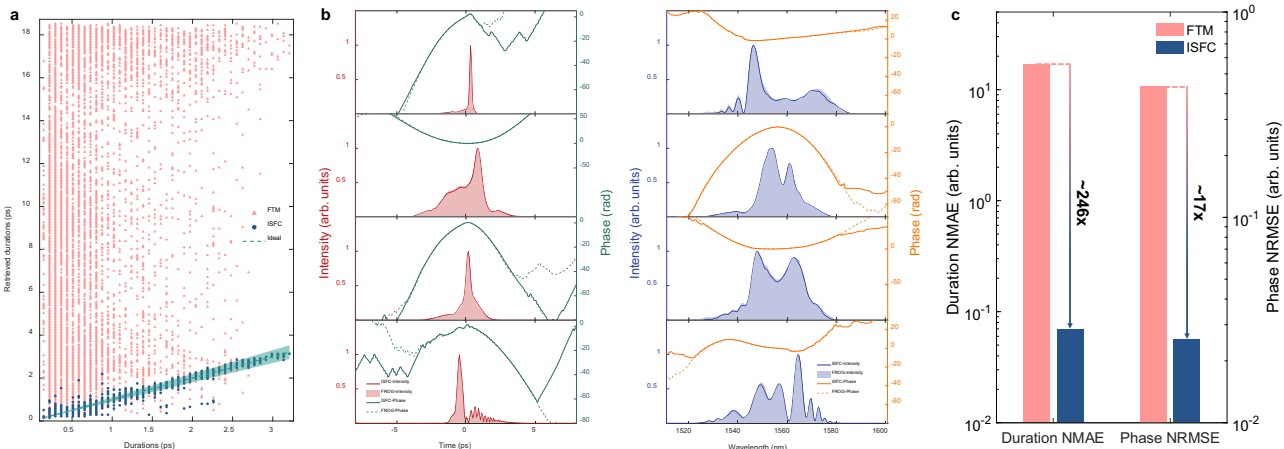

**Fig. 5 | Comparison of FTM and ISFC for fs pulse reconstruction in experiments. a** Pulse duration retrieval results of FTM and ISFC, where the dashed line and shadow area respectively denote the ideal situation and the 10% error range. **b** Accurate pulse reconstruction of ISFC generalizes to the short pulse (top row),

pulses with different signs of chirp (middle rows), and the pulse with a sophisticated shape (bottom row). **c** Statistics comparison between FTM and ISFC. The duration NMAE of ISFC and FTM are -0.07 and 17.12, respectively. The phase NRMSE of ISFC and FTM are 0.0256 and 0.4337 respectively.

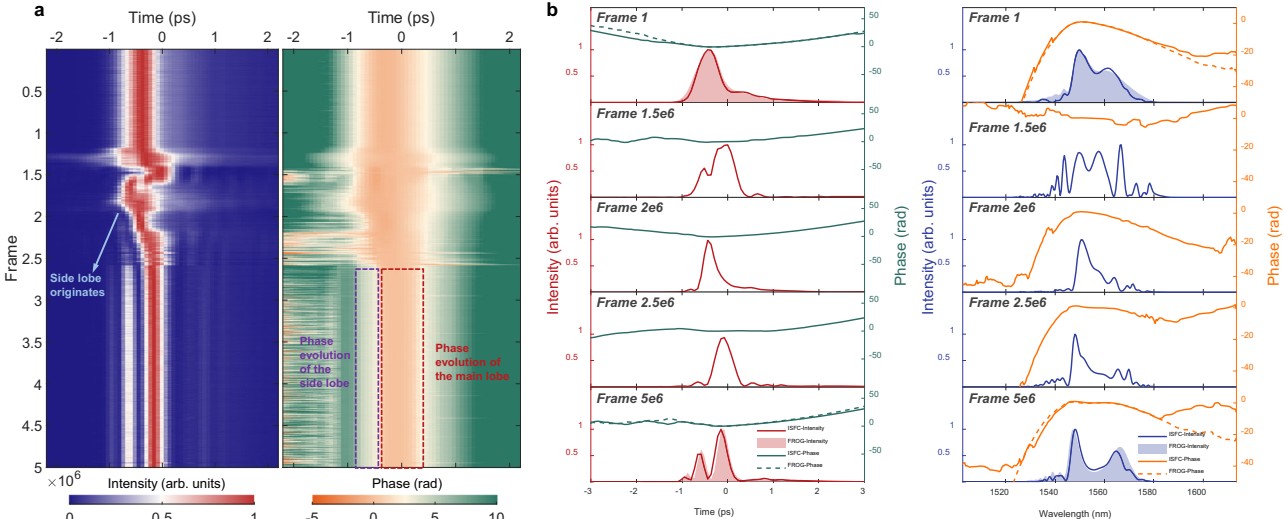

**Fig. 6 | Switching dynamics of the programmable spectral filter resolved by ISFC in single shot. a** The intensity (left) and phase (right) evolution dynamics in a switching process of the programmable spectral filter resolved by ISFC in single

shot under the sampling rate of 3.13 GSa/s. **b** Reconstructed pulses of the start frame, 3 middle frames, and the final frame, where the reconstructed start and final pulses align well with the FROG measurements.

branch to FROG for label collection is 4.12 mW, corresponding to the pulse energy of -127.2 pJ. The shortest nonlinear length of the FROG branch is -0.87 m, and the length of the fiber jumper connecting one output arm of the optical coupler and the spatial collimator is -0.4 m. In the FROG branch, the shortest nonlinear length is still comparable to the actual length of fiber links, but the nonlinearities are rather weaker than in the LSSI branch. The LSSI branch generating inputs for ISFC and the FROG branch generating labels for ISFC experience distinctly different strengths of nonlinearities. As a result, the demonstrated experimental results of ISFC in pulse reconstruction are inferior to its simulation results. One piece of evidence that validates the deduction is that the outliers in the retrieved pulse durations shown in Fig. 5a substantially concentrate on shorter pulses suffering from more severe nonlinearities under close pulse energies. On the other hand, measurement errors are inevitable in label collection via FROG, but the labels in simulations are perfectly correct. Noise in experiments is also a non-negligible factor in performance deterioration, which is

validated by evaluating the pulse reconstruction performance under different noise levels in simulations (see Supplementary Information).

The LSSI-based ISFC completely relies on linear effects, thus making it suitable for realizing single-shot full-field characterization over low-energy fs pulses. Linear systems are generally limited by noise level. As the quantization noise induced performance deterioration is comparatively small in the simulation results shown in Supplementary Fig. 1, we suppose the main noise is introduced in photodetection in experiments. The bandwidth and noise equivalent power of EOT ET-3500F is >15 GHz and 28 $pW/\sqrt{Hz}$ respectively. Thus, the noise power of photodetection reaches 3.43 uW. According to numerical simulations, ISFC manifests decent robustness against noise, and its pulse reconstruction performance is acceptable under the signal-to-noise ratio of 0 dB with the duration NMAE of -0.08, as shown in Supplementary Fig. 1. Hence, the signal power prior to photodetection should be larger than 3.43 uW. The insertion loss of the experimental LSSI is measured to be -8.49 dB, and the pulse power into the LSSI should be

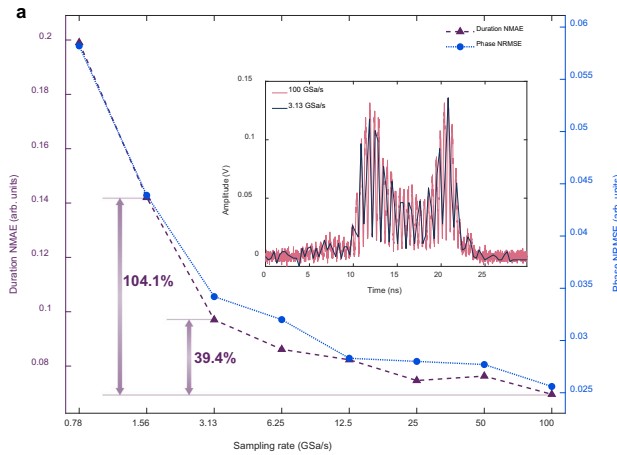

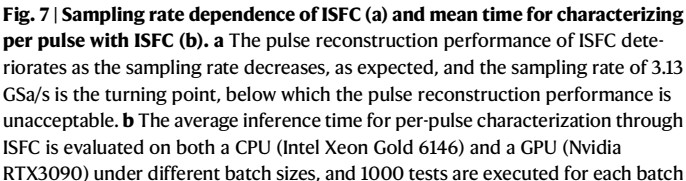

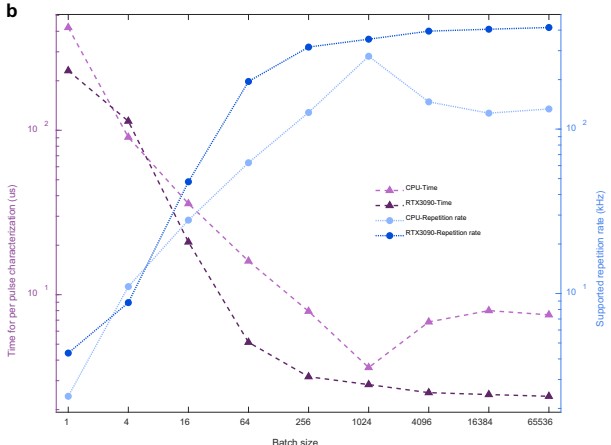

**Fig. 7 | Sampling rate dependence of ISFC (a) and mean time for characterizing per pulse with ISFC (b). a** The pulse reconstruction performance of ISFC deteriorates as the sampling rate decreases, as expected, and the sampling rate of 3.13 GSa/s is the turning point, below which the pulse reconstruction performance is unacceptable. **b** The average inference time for per-pulse characterization through ISFC is evaluated on both a CPU (Intel Xeon Gold 6146) and a GPU (Nvidia RTX3090) under different batch sizes, and 1000 tests are executed for each batch size to obtain the average inference time. Due to the compute unified devices architecture (CUDA) acceleration, the advantage of the GPU on average inference time over the CPU is magnified as the batch size increases. The average inference time for per-pulse characterization on the GPU gradually converges to 2.41 us, corresponding to the supported highest repetition rate of ~414 kHz for online single-shot full-field characterization.

above ~24.23 uW. Therefore, the demonstrated LSSI-based ISFC experimental system theoretically supports the pulse energy of merely ~748 fJ for realizing decent single-shot full-field characterization over fs pulses, but the required pulse energy of FROG to perform single-shot measurement is at the microjoule level[16,20]. Moreover, by replacing the current photodetector with a high-sensitivity avalanche photodiode, the minimal pulse energy of the LSSI-based ISFC to achieve single-shot measurement may be further reduced.

For ultraviolet wavelength and X-ray band, high-efficiency nonlinearities generation is very challenging due to the strong absorption of conventional nonlinear media on these ultrashort wavelength bands[25–27]. As a result, characterizing the ultrashort-wavelength fs pulses via FROG or SPIDER is difficult. The LSSI-based ISFC inherently adapts to different wavelengths as long as the corresponding dispersive medium is available, which provides a promising solution to characterize low-energy ultraviolet-wavelength fs pulses in single shot. The implementation of LSSI is flexible enough to support its utilization in various wavelengths. For instance, the spatial Michelson interferometer and diffraction gratings can also constitute the LSSI. Although dispersion is easier to obtain than nonlinearities for shorter wavelengths, the challenge of measuring ultraviolet fs pulses and even attosecond pulses via LSSI-based ISFC still lies in dispersion. LSSI requires the dispersive component providing large group velocity dispersion over a wide spectral range, covering the spectra of the pulses under measurement. Finding or designing the suitable dispersive component becomes a challenge. Certainly, the required dispersion amount can be reduced as the sampling rate increases. The LSSI-based ISFC can be applied to measure even shorter fs pulses accurately as shown in Supplementary Fig. 5. The shortest pulse, which can be characterized by ISFC, is confined by the bandwidth of the dispersive medium in the LSSI. Therefore, given an ultra-large-bandwidth dispersive medium, single-shot full-field characterization over attosecond pulses via the LSSI-based ISFC can be expected.

To summarize, we demonstrate single-shot full-field characterization over the fs pulse train with a repetition rate of ~32.4 MHz through LSSI-based ISFC. Benefiting from the linear effects reliance of LSSI, the required pulse energy for single-shot full-field characterization via ISFC is reduced to ~296.6 pJ. Compared to the conventional FTM, ISFC allows to use smaller spectral shear values for accurate pulse reconstruction, thereby significantly reducing the required sampling

rate of the acquisition device. As a result, the switching dynamics of the programmable spectral filter are resolved by ISFC under a readily available sampling rate of 3.13 GSa/s. Moreover, ISFC is more robust against both quantization noise and random noise than FTM in numerical simulations. We believe LSSI-based ISFC is highly competitive in single-shot full-field characterization over fs pulse trains, especially in weak ultraviolet fs pulses and even attosecond pulses characterization.

## Methods
### Numerical simulation of LSSI
In numerical simulations, the temporal resolution is 10 fs and the time window is 20 ns. To generate pulses with asymmetric profiles, the fs pulse under characterization is synthesized by two randomly chirped Gaussian pulses, which can be expressed as

$$E_1(t) = A_1(t)e^{-\left(\frac{1+jC_{11}}{2}\left(\frac{t}{T_{01}}\right)^2 + \frac{jC_{12}}{6}\cdot\left(\frac{t}{T_{01}}\right)^3\right)}, \quad (6)$$

$$E_2(t) = A_2(t)e^{-\left(\frac{1+jC_{21}}{2}\left(\frac{t}{T_{02}}\right)^2 + \frac{jC_{22}}{6}\cdot\left(\frac{t}{T_{02}}\right)^3\right)}, \quad (7)$$

$$E(t) = E_1(t) + \alpha \cdot E_2(t - \tau). \quad (8)$$

$C_{11}$, $C_{12}$, $C_{21}$, $C_{22}$ are random chirp parameter of two sub-pulses, $T_{01}$ and $T_{02}$ are random pulse durations of two sub-pulses. A random small delay $\tau$ is introduced between the two sub-pulses to create the asymmetric profile and $\alpha$ is the random attenuation factor. As a result, $E_1(t)$ serve as the main part and $E_2(t - \tau)$ is the secondary part of the synthesized fs pulse.

The fs pulse under characterization is centered at 1030 nm and first goes through the interferometer. Datasets with different delays ranging from 1 ps to 5 ps are generated. During the generation of the data with delay jitter, the delay is jittered randomly within the allowed maximal jitter magnitude for each pulse. By combining the raw pulse and its delayed replica, the spectral fringe patterns can be obtained. Then, DFT is utilized to introduce a spectral shear between the two pulses and map the spectral fringes into time simultaneously. DFT is

performed by a 3-km SM980 fiber with the group velocity dispersion of about -47.8 ps/nm/km at 1030 nm. We ignore nonlinearities in the fiber for faster data generation, and the dispersion can be modeled by a spectral phase filter as follows

$$E_{DFT}(\omega) = \mathcal{F}(E(t) + E(t - \Delta t)) \cdot e^{j\left(\frac{\beta_2 L}{2} \cdot \omega^2 + \frac{\beta_3 L}{6} \cdot \omega^3\right)}. \qquad (9)$$

$\beta_2$, $\beta_3$ are the group velocity dispersion and TOD of the fiber respectively and $L$ is the fiber length. $E_{DFT}(\omega)$ is the spectral field after DFT. The shear value calculated via Eq. (1) is -9.85 GHz. After signal acquisition, including photodetection, and bandwidth limit (a low-pass filter with a bandwidth of 20 GHz and super-Gaussian rolling off), the temporal interferogram containing full-field information of the fs pulse under characterization is obtained. We generate 60,000 sets of data as the basic simulation dataset, and the duration of 60,000 pulses ranges from 50 fs to 1.4 ps. Other variant simulation datasets can be readily obtained by downsampling the temporal interferogram of the basic simulation dataset with different sampling rates, quantizing it with different resolutions, and adding different levels of Gaussian random noise to it. 54,000 sets of data are used for training, 3000 sets of data are used for validation, and the rest 3000 sets of data are used for testing.

### Experimental details and ISFC training

The programmable fs laser is built using a homemade C-band fs laser with a repetition rate of 32.4 MHz, followed by a programmable spectral filter (Finisar WaveShaper 4000B). A commercial FROG (MesaPhotonics FROGScan Ultra) is used for label collection. The temporal resolution of the FROG is set to -70 fs with 256 scanning points. The FROG updates the measurement result every -7 s, and the updating time becomes larger when more scanning points are used. The claimed group velocity dispersion of the DCF is -340 ps/nm. The dispersion amount has to satisfy the far-field Fraunhofer condition described by the inequation below for realizing DFT, where $t_{FWHM}$ is the temporal width. $\beta_2$ and $L$ are the group velocity dispersion and length of the dispersive medium. It sets a lower bound for the dispersion amount and the wider pulses require larger group velocity dispersion to satisfy far-field Fraunhofer condition. On the other hand, when time delay is fixed, the dispersion determines the shear frequency and too small dispersion causes the very high shear frequency. As a result, the demands on the sequential acquisition system become quite high and are very hard to be met.

$$\frac{t_{FWHM}^2}{2\pi\beta_2 L} \ll 1 \qquad (10)$$

The dispersed signal reaches nanosecond-level width in time domain (see Supplementary Fig. 3) and it can be readily acquired by the high-speed real-time acquisition system. The bandwidth of the photodetector (EOT ET-3500F) is >15 GHz. A real-time oscilloscope (Tektronix DPO70000SX Series) is used for interferogram acquisition at the sampling rate of 100 GSa/s and the bandwidth of 33 GHz. 70,000 samples are preserved in the experiments for training ISFC, with 63000 samples for training and the rest 7000 samples for testing. Each pulse in the dataset is created by imposing a random phase curve generated by the WaveShaper. The delay of the Mach-Zehnder interferometer is -5.8 ps calibrated with an amplified spontaneous emission source, and the group velocity dispersion and the TOD are respectively --339.65 ps/nm and --1.89 ps³ calibrated via temporal-to-spectral mapping built by DFT (see Supplementary Information).

The consecutive temporal interferograms are first segmented into separate frames with a frame length of 30 ns. Cross-correlation is applied among different frames corresponding to different fs pulses to align these frames horizontally. The input dimension of the fully-connected neural network in ISFC is determined by the sampling rate, and the input dimension is 93 under the sampling rate of 3.13 GSa/s. Three hidden layers with dimensions of 512, 256, and 512, respectively are used. The output dimension is 512 representing the real and imaginary parts of the pulse. Layer normalization is used to alleviate internal covariate shifts between two layers and outperforms batch normalization in training[31,32]. The activation function is ReLU, and dropout is applied to alleviate overfitting. The fully-connected neural network is implemented using PyTorch, and key hyperparameters (e.g., the learning rate of 2.85e-4, training epochs of 500, the dropout ratio of 4.089e-2, and the batch size of 128) are optimized by Optuna. The optimizer is Adam and the cosine annealing learning rate strategy is applied for better convergence.

## Data availability

The minimum data for validation have been deposited in Code Ocean (https://codeocean.com/capsule/2832276/tree/v1). The entire data is available from the corresponding author upon request without any commercial interest.

## Code availability

Core codes are available on Code Ocean (https://codeocean.com/capsule/2832276/tree/v1). Other codes used in this manuscript are available from the corresponding author upon request without any commercial interest.

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

## Acknowledgements

This work was supported by the National Natural Science Foundation of China (62205199 to G. Pu, 62227821 to L. Yi, 62025503 to L. Yi).

## Author contributions

G.P. conceived of the idea, designed and performed the experiments under the supervision of L.Yi. C.L. assisted in the experiments. All authors discussed the results and contributed to writing the manuscript.

## Competing interests

The authors declare no competing interests.
