## [Transparent Peer Review file · Nature Communications]

Intelligent single-shot full-field characterization over femtosecond pulses

Corresponding Author: Professor Lilin Yi

Version 0:

Reviewer comments:

Reviewer #1

(Remarks to the Author)

This manuscript reports on a novel method for single-shot complete characterization of high-repetition-rate femtosecond pulse trains. In particular, the authors combine linear spectral shearing interferometry (LSSI) with a machine learning technique to provide an innovative solution to this challenging task. Although spectral shearing interferometry alone enables single-shot complete characterization in principle, its reliance on precise calibration and its poor robustness to perturbations have hindered practical applications. The authors prove that the two inherent drawbacks of LSSI can be overcome by a well-trained fully-connected neural network in both numerical simulations and experiments. Interestingly, the machine learning model helps to lower the requirements on the sampling device. The proposed method is promising to become a widely used characterization method for femtosecond pulses. I think this work is of considerable interest to the community and worth publishing in Nature Communications, but I recommend the authors consider the following points to improve the manuscript:

1. Why does the output of the neural network need to be arranged in an interleaved format between the real and imaginary parts? How does the performance compare when simply concatenating intensity and phase instead? Moreover, since the model is currently trained in a supervised manner, could the authors clarify whether its performance might be improved with unsupervised or self-supervised training?
2. The authors mention that the temporal resolution of the FROG is 70 fs, but the duration MAE on the experimental test set of ISFC shown in Fig. 5 is 41.32 fs, which is even smaller than the temporal resolution of the FROG, could the authors explain this discrepancy?
3. Figure 6 shows the switching dynamics of the programmable spectral filter as resolved by ISFC in single shot. How can the authors ensure that the reconstructed intermediate frames are accurate? Are there any validation methods or ground-truth comparisons?
4. I suggest that the authors provide a more thorough discussion on the required sampling rate. Could it be further reduced? What are the theoretical and practical limitations that constrain this rate?

(Remarks on code availability)

Reviewer #2

(Remarks to the Author)

The Authors present a fully connected neural network-based linear spectral shearing interferometry (LSSI) technique for single-shot, full-field characterization of femtosecond pulses. This approach enables real-time pulse characterization without the use of a spectrometer and, importantly, allows for the measurement of weak femtosecond pulses. By reconstructing ultrafast pulses without relying on nonlinear effects, the method provides a direct and robust means of characterization. The manuscript has novelty and is clearly written, and addresses an important issue in ultrafast optics. However, I would like the authors to address a few of major and minor issues before the decision.

1. Key application related

1.1 The authors emphasize that the key application of the proposed technique is the characterization of weak ultraviolet femtosecond pulses, and even attosecond pulses at high repetition rates. Has there been any previous research demonstrating the applicability of LSSI methods to weak ultraviolet or attosecond pulse measurements?

1.2 Is it feasible to induce a precise linear spectral shearing effect in the UV and EUV ranges of femtosecond pulses? A discussion of the practical challenges and limitations would strengthen the manuscript.

1.3 To my understanding, direct gating methods and FROG-based techniques have been previously employed for such applications. Could the authors clarify what specific advantages their proposed method offers compared to these established approaches?

1.4 It would be better to show the LSSI system (optical layout) used in these experiments. (In supplement)

2. Complete Characterization of Ultrafast Pulses

2.1 In this work, the authors trained the LSSI model and performed experimental demonstrations using pulses with durations above 40 fs. However, ultrafast pulses are typically defined as having pulse widths below 10 fs. (Ti:Sapphire laser) Is it feasible to demonstrate the method to this short(10 fs) pulse regime?

2.2 What is the minimum pulse duration that can be fully characterized using the LSSI-based method, both experimentally and with the AI-enhanced model? Please clarify the lower limit of applicability.

2.3 Defining the critical group delay dispersion (GDD) limit is essential for establishing the reliability of any ultrafast pulse characterization technique. As pulse durations become shorter, GDD and higher-order phase effects (e.g., third-order dispersion) become increasingly significant. In this manuscript, no critical GDD limit is presented, partly because the pulses used are relatively long. Could the authors test their approach using the shortest pulse duration available from their laser system to evaluate and define the critical GDD limit?

3. Training of the AI Model

3.1 The authors briefly describe the training procedure in the Methods section. However, providing more detailed information about the model (such as the optimizer used, the number of layers, the number of epochs, and other relevant hyperparameters) would greatly enhance the readers' understanding.

3.2 The choice of a fully connected architecture is mentioned but not justified. Could the authors explain why this architecture was selected and what advantages it offers compared to alternatives (e.g., CNNs, RNNs, or hybrid models)?

3.3 The value of $\Delta\omega$ is expected to influence both the experimental results and the training performance of the AI model. Was this parameter fixed during the training process? If so, please clarify how it was chosen.

3.4 More detailed information about the preparation of training data would also improve the clarity of the manuscript. Could the authors describe in greater detail the data generation and preprocessing steps used to train the AI model?

4. Comparison between FTM and ISFC

4.1 Figures 2 and 5 present a comparison between the FTM and ISFC methods. The results indicate that ISFC is highly effective for LSSI-based pulse reconstruction, whereas the FTM method appears to perform very poorly. This raises concerns, as it is difficult to accept that FTM-based LSSI could be useful at all prior to the introduction of the AI-based approach. Could the authors elaborate on this point? Does the FTM method inherently suffer from such low accuracy, or might this comparison exaggerate its limitations? If the former is true, is it then an appropriate baseline for comparison?

4.2 In Figure 5, the authors compare ISFC with FROG. Why was a more detailed statistical analysis of the FROG results not performed? Such the analysis helps the robustness and generalizability of the proposed method.

5. Switching Dynamics

5.1 Figure 6 presents the switching dynamics of the programmable spectral filter. The results show excellent agreement between ISFC and FROG, which strongly supports the validity of the characterization. However, it would be helpful if the authors could clarify in which specific applications or scenarios such switching characteristics could be practically utilized.

5.2 What is the shortest pulse duration for which this switching dynamic can be reliably demonstrated? Please provide a discussion of the limits.

(Remarks on code availability)

Reviewer #3

(Remarks to the Author)

In the manuscript, the authors demonstrate the intelligent single-shot full-field characterization of femtosecond pulses based on linear spectral shearing interferometry. Since it is a linear process, the work significantly reduced laser pulse energy compared with methods that employ nonlinear process for laser pulse measurements such as SPIDER and FROG. By employing machine learning, the work obviously reduced the affect by noise. The manuscript is well written and the results are interesting. Before I recommend the manuscript accept by Nature Communications, I would like to ask the authors to consider my concerns below for further improvement of the manuscript.

1. In the manuscript, the signal is detected employing photo detector and oscilloscope. But is the sampling speed enough since the measured pulse is femtosecond and the dispersed pulse duration is in ps.
2. What is the pulse measurement accuracy and how short the pulse can be measured? For even shorter pulses, GDD, TOD and FOD are all important for the pulse measurement, but the work only measured up to TOD.
3. In the manuscript, the authors claimed that the method can measure attosecond pulses. Is it possible since the measurement accuracy requires is quite high. How reliable is it? Is there any suitable dispersion materials can provide sufficient dispersion?
4. What is the requirement on training data accuracy, which is quite important for characterizing ultrashort laser pulses with high reliability?
5. How much amount dispersion is required for the measurement, considering the temporal resolution of detector and oscilloscope.
6. In the work, sampling rate is crucial thus high repetition rate is required. What is the lowest repetition rate that can be accepted. What is the limitations of the method.
7. Can the method measure electric waveform of the laser pulses, which really requires high accuracy and resolution, and there are works have reported such as TIPTOE, FROG and SPIDER.

(Remarks on code availability)

Version 1:

Reviewer comments:

Reviewer #1

(Remarks to the Author)

I am satisfied with the revised version and can recommend acceptance.

(Remarks on code availability)

Reviewer #2

(Remarks to the Author)

The authors have provided a comprehensive and well-prepared rebuttal that addresses the majority of the reviewers' comments in a clear and scientifically rigorous manner. The revised manuscript has been substantially improved both in clarity and completeness.

The proposed 'LSSI-based intelligent single-shot full-field characterization (ISFC)' represents a technically innovative approach that combines linear spectral shearing interferometry with data-driven reconstruction. The idea of employing a neural network to circumvent calibration instabilities and high-order dispersion issues is novel and potentially impactful. However, I still have one concern regarding short (~10 fs level) pulse laser with conventional dispersive medium that causing group delay dispersion. (Practical point of view and precision.)

The authors claim that ISFC can characterize pulses as short as 8 fs based on simulations including TOD. However, in practice, broadband femtosecond pulses inevitably experience strong dispersion and spectral phase distortion through the DCF or grating used in LSSI. Such high-order dispersion leads to non-linear temporal-to-spectral mapping that may not be easily compensated by data-driven approaches trained on limited ranges of β_2 and β_3 . Therefore, it remains unclear whether the proposed technique can practically characterize sub-20 fs pulses under realistic experimental conditions. The authors are encouraged to discuss this limitation more concretely or provide preliminary experimental verification with broader-bandwidth pulses.

(Remarks on code availability)

Reviewer #3

(Remarks to the Author)

The authors as carefully respond to my comments and make revisions to the manuscript. I agree that the manuscript can be

accepted now.

(Remarks on code availability)

We sincerely thank the reviewers for their careful reviewing of the manuscript entitled “Intelligent single-shot full-field characterization over femtosecond pulses” by Guoqing Pu, Chao Luo, Weisheng Hu, and Lilin Yi. We appreciate their invaluable comments. We have revised the manuscript according to the comments of the reviewers. The revised manuscript is provided. For your convenience, we provide this response letter with the reviewer’s comments in *italic*, with our revisions underlined and **highlighted in yellow**. In the following, we answer the comments raised by the reviewer in detail.

Comments from Reviewer #1:

This manuscript reports on a novel method for single-shot complete characterization of high-repetition-rate femtosecond pulse trains. In particular, the authors combine linear spectral shearing interferometry (LSSI) with a machine learning technique to provide an innovative solution to this challenging task. Although spectral shearing interferometry alone enables single-shot complete characterization in principle, its reliance on precise calibration and its poor robustness to perturbations have hindered practical applications. The authors prove that the two inherent drawbacks of LSSI can be overcome by a well-trained fully-connected neural network in both numerical simulations and experiments. Interestingly, the machine learning model helps to lower the requirements on the sampling device. The proposed method is promising to become a widely used characterization method for femtosecond pulses. I think this work is of considerable interest to the community and worth publishing in Nature Communications, but I recommend the authors consider the following points to improve the manuscript:

Comment 1:

Why does the output of the neural network need to be arranged in an interleaved format between the real and imaginary parts? How does the performance compare when simply concatenating intensity and phase instead? Moreover, since the model is currently trained in a supervised manner, could the authors clarify whether its performance might be improved with unsupervised or self-supervised training?

Response 1:

We sincerely thank the reviewer for highly praising our works and raising the questions. We design the neural network to output both real and imaginary parts to realize full-field reconstruction over fs pulses with only one network instead of two (e.g., one for predicting the magnitude and the other for prediction the phase). As shown in Table R1, we have tried to train with the concatenating intensity and phase, but the performance is inferior. This may be caused by the large value difference between the magnitude and phase as shown in Fig. R1. The discontinuous point between the magnitude and phase also enhances the training difficulty.

Supervised learning relies on numerous labelled data. As a result, for an identical task, the performance of supervised learning is usually better than unsupervised or self-supervised learning. However, obtaining numerous labelled data can be costly in experiments. Therefore, to realize unsupervised or self-supervised learning with decent performance is valuable and we are working on it. We have revised the manuscript accordingly as follows.

Table R1. Different types of labels and their performance on the experimental test set

	Real & Imag interleaved	Real & Imag concatenated	Magnitude & Phase concatenated
Duration MAE	41.32 fs	42.43 fs	196.92 fs
Phase NRMSE	0.0251	0.0253	0.0486

Figure R1. One magnitude & phase concatenated label

Revision 1:

The last paragraph of ISFC (Intelligent single-shot full-field characterization) over fs pulses via LSSI

ISFC is designed to output the interleaved real and imaginary parts of the temporal pulse, and the temporal field can be readily reconstructed. **We found that the interleaved sequencing outperformed concatenating the intensity and phase.** Through the Fourier transform, the spectral field of the pulse can also be reconstructed. It is equivalent for ISFC to predict the spectral field first and then obtain the temporal field via the Fourier transform.

Comment 2:

The author mention that the temporal resolution of the FROG is 70 fs, but the duration MAE on the experimental test set of ISFC shown in Fig. 5 is 41.32 fs, which is even smaller than the temporal resolution of the FROG, could the authors explain this discrepancy?

Response 2:

We thank the reviewer for raising this question. 41.32 fs is the mean absolute error (MAE) of pulse durations on the experimental test set. Because ISFC accurately retrieves the pulse duration for most pulses in the test set as shown in Fig. R1, the duration MAE is a statistical characteristic and it could be smaller than the temporal resolution of 70 fs.

Comment 3:

Figure 6 shows the switching dynamics of the programmable spectral filter as resolved by ISFC in single shot. How can the authors ensure that the reconstructed intermediate frames are accurate? Are there any validation methods or ground-truth comparisons?

Response 3:

We thank the reviewer for prompting this question. Unfortunately, there is no way to realize full-field characterization over fs pulse train at MHz frame rate. Therefore, there is no ground truth to cross validated the ISFC retrieved switching dynamics. However, the switching dynamics in Fig. 6 is deduced to be correct for the following two reasons. First, the reconstructed start and final pulses align well with the FROG measurements in both temporal and spectral domains. Second, the evolutionary process is consecutive as evidenced in the temporal intensity evolution. We have revised the manuscript accordingly as follows.

Revision 3:

The fourth paragraph of Experimental validation of ISFC over fs pulses via LSSI

Figure 6b shows the reconstructed pulses of the start frame, the 3 middle frames, and the final frame. Unfortunately, since there is no way to realize full-field characterization over fs pulse train at MHz frame rate, the ISFC resolved full-field switching dynamics cannot be cross validated. However, the validity of the resolved switching dynamics can be guaranteed by for the following two reasons. First, the reconstructed start and final pulses align well with the FROG measurements in both temporal and spectral domains as shown in Fig. 6b. Second, the evolutionary process is consecutive as evidenced in the temporal intensity evolution.

Comment 4:

I suggest that the authors provide a more thorough discussion on the required sampling rate. Could it be further reduced? What are the theoretical and practical limitations that constrain this rate?

Response 4:

We thank the reviewer for raising this question. An empirical deduction drawn from the simulation (see Supplementary Information) and experimental results indicates that the minimal sampling rate supported by ISFC lies between the actual shear value and its corresponding Nyquist sampling rate. Though we do not suppose ISFC violates the Nyquist sampling theorem. Because the shear value remains unchanged in simulations and it can be learnt by ISFC through numerous data. When the sampling rate is below the Nyquist sampling rate, ISFC automatically pads the missing but deterministic data points during computation. The shear value in experiments varies due to the environmental disturbances. However, the variation is rather small as the fringe frequencies of different temporal interferograms are all very close to the calibrated shear value of ~2.13 GHz. As a result, the pulse reconstruction performance of ISFC is acceptable under the sampling rate of 3.13 GSa/s. On the other hand, the performance drop induced by using the sampling rate below the Nyquist sampling rate is quite obvious in both simulations and experiments.

Revision 4:

The fifth paragraph of Experimental validation of ISFC over fs pulses via LSSI

The pulse retrieval performance sharply deteriorates when further reducing the sampling rate to 1.56 GSa/s, as evidenced by the duration NMAE drastically increasing by 104.1% compared to the sampling rate of 100 GSa/s. An empirical deduction drawn from the simulation and experimental results indicates that the minimal sampling rate supported by ISFC lies between the actual shear value and its corresponding Nyquist sampling rate (see Supplementary Information).

Comments from Reviewer #2:

The Authors present a fully connected neural network-based linear spectral shearing interferometry (LSSI) technique for single-shot, full-field characterization of femtosecond pulses. This approach enables real-time pulse characterization without the use of a spectrometer and, importantly, allows for the measurement of weak femtosecond pulses. By reconstructing ultrafast pulses without relying on nonlinear effects, the method provides a direct and robust means of characterization. The manuscript has novelty and clearly written, and addresses an important issue in ultrafast optics. However, I would like the authors to address a few of major and minor issues before the decision.

Comment 1:

1. Key application related

1.1 The authors emphasize that the key application of the proposed technique is the characterization of weak ultraviolet femtosecond pulses, and even attosecond pulses at high repetition rates. Has there been any previous research demonstrating the applicability of LSSI methods to weak ultraviolet or attosecond pulse measurements?

1.2 Is it feasible to induce a precise linear spectral shearing effect in the UV and EUV ranges of femtosecond pulses? A discussion of the practical challenges and limitations would strengthen the manuscript.

1.3 To my understanding, direct gating methods and FROG-based techniques have been previously employed for such applications. Could the authors clarify what specific advantages their proposed method offers compared to these established approaches?

1.4 It would be better to show the LSSI system (optical layout) used in these experiments. (In supplement)

Response 1.1:

We sincerely thank the reviewer for the general comments and proposing the questions. To the best of our knowledge, there is no demonstration of weak ultraviolet or attosecond pulse measurements through LSSI. Due to the lack of ultraviolet fs lasers and attosecond pulse sources in our lab, we also fail to demonstrate it. Technically speaking, dispersion and interferometry is all that LSSI needs to achieve

single-shot full-field characterization over weak ultraviolet fs pulse. Attosecond pulses can be measured by spectral shear interferometry [Quéré F, Itatani J, Yudin G L, et al. Attosecond spectral shearing interferometry[J]. *Physical Review Letters*, 2003, 90(7): 073902] and, therefore, attosecond pulses can also be measured by LSSI.

Response 1.2:

We thank the reviewer for giving suggestions. Though dispersion is usually easier to obtain than nonlinearities, we suppose the real challenge still lies in dispersion. LSSI requires the dispersive component providing large group velocity dispersion over wide spectral range, covering the spectra of the pulses under measurement. The required dispersion amount can be reduced as the sampling rate increases. Overall, we believe it is feasible to measure UV and even EUV fs pulses via LSSI. We have revised the manuscript accordingly as follows.

Revision 1.2:

The third paragraph of Discussion

For ultraviolet wavelength and X-ray band, high-efficiency nonlinearities generation is very challenging due to the strong absorption of conventional nonlinear media on these ultrashort wavelength bands^{Error! Reference source not found.}. As a result, characterizing the ultrashort-wavelength fs pulses via FROG or SPIDER is difficult. The LSSI-based ISFC inherently adapts to different wavelengths as long as the corresponding dispersive medium is available, which provides a promising solution to characterize low-energy ultraviolet-wavelength fs pulses in single shot. The implementation of LSSI is flexible enough to support its utilization in various wavelengths. For instance, the spatial Michelson interferometer and diffraction gratings can also constitute the LSSI. Although dispersion is easier to obtain than nonlinearities for shorter wavelengths, the challenge of measuring ultraviolet fs pulses and even attosecond pulses via LSSI-based ISFC still lies in dispersion. LSSI requires the dispersive component providing large group velocity dispersion over a wide spectral range, covering the spectra of the pulses under measurement. Finding or designing the suitable dispersive component becomes a challenge. Certainly, the required dispersion amount can be reduced as the sampling rate increases. The LSSI-based ISFC can be applied to measure even shorter fs pulses accurately as shown in Fig. S5. The shortest pulse, which can be characterized by ISFC, is confined by the bandwidth of the dispersive medium in the LSSI. Therefore, given an ultra-large-bandwidth dispersive medium, single-shot full-field characterization over attosecond pulses via the LSSI-based ISFC can be expected.

Response 1.3:

We thank the reviewer for raising the question. The proposed LSSI-based ISFC holds two advantages over conventional approaches. **First, LSSI-based ISFC can measure weak fs pulses in single shot.** Direct gating method and FROG-based techniques certainly can achieve full-field characterization over fs pulses. However, as we summarized in the introduction, all these techniques rely on nonlinearities thereby posing an energy requirement on pulses under measurement. The situation becomes even severer when it comes to single shot characterization. For instance, the required pulse energy of FROG to

perform single-shot measurement is at the microjoule level. On the other hand, the proposed LSSI-based ISFC realize single-shot full-field characterization over fs pulses with the energy of ~ 300 pJ.

Second, the proposed LSSI-based ISFC delivers direct and fast pulse reconstruction. Many conventional approaches (e.g., FROG-based techniques, D-scan) requires iterative phase retrieval. As a result, it is quite slow for them to update the measurement results. FROG we used in experiments (MesaPhotonics FROGScan) update the measurement results every ~ 7 s under the FROG trace size of 256×256 and it could be even slower when we choose larger FROG trace size in the setting (e.g., 512×512 , 1024×1024). However, LSSI-based ISFC can reconstruct the pulse in 2.41 μ s averagely and it represents the supported highest repetition rate of allowing online single-shot full-field characterization via ISFC theoretically reaches ~ 414 kHz.

Response 1.4:

We thank the reviewer for the advice. However, the entire optical layout is clearly shown in the upper part of Fig. 4 and we paste it below. The programmable fs laser in the optical layout is built using a homemade C-band fs laser with a repetition rate of 32.4 MHz, followed by a programmable spectral filter (Finisar WaveShaper 4000B). Then, a commercial FROG (MesaPhotonics FROGScan Ultra) is used for label collection. The LSSI system contains a fiber MZI and a spool of DCF for performing DFT. The fiber MZI consists of two fiber couplers and a fiber tunable delay.

Figure R2. The optical layout of LSSI system.

Comment 2:

2. Complete Characterization of Ultrafast Pulses

2.1 *In this work, the authors trained the LSSI model and performed experimental demonstrations using pulses with durations above 40 fs. However, ultrafast pulses are typically defined as having pulse widths below 10 fs. (Ti:Sapphire laser) Is it feasible to demonstrate the method to this short(10 fs) pulse regime?*

2.2 *What is the minimum pulse duration that can be fully characterized using the LSSI-based method, both experimentally and with the AI-enhanced model? Please clarify the lower limit of applicability.*

2.3 *Defining the critical group delay dispersion (GDD) limit is essential for establishing the reliability of any ultrafast pulse characterization technique. As pulse durations become shorter, GDD and higher-order phase effects (e.g., third-order dispersion) become increasingly significant. In this manuscript, no critical GDD limit is presented, partly because the pulses used are relatively long.*

Could the authors test their approach using the shortest pulse duration available from their laser system to evaluate and define the critical GDD limit?

Response 2.1:

We thank the reviewer for raising the question. Due to the capacity of our homemade fs laser, we feel sorry we cannot experimentally demonstrate using the LSSI-based ISFC to measure pulses with durations below 10 fs. However, we validate characterization over even shorter fs pulses with the LSSI-based ISFC in simulations and the corresponding pulse reconstruction results are shown in Fig. S5. The average duration MAE and the average phase RMSE over 6000 pulses in the test set are 0.01 fs and 0.071 rad respectively, manifest rather good pulse reconstruction. Actually, the bandwidth of the dispersive medium in the LSSI determines the shortest pulse which can be characterized by ISFC. Therefore, it is feasible to apply the LSSI-based ISFC to measure very short pulses.

Revision 2.1:

Add D. Characterization over shorter fs pulses with LSSI-based ISFC in Supplementary Information

D. Characterization over shorter fs pulses with LSSI-based ISFC

To validate the ability of the proposed LSSI-based ISFC in even shorter fs pulses characterization, we generate a shorter-fs-pulse dataset in numerical simulations, which contains 60000 pulses with durations ranging from 8 fs to 50 fs. Due to the wide bandwidth, the fibre length for performing DFT can be reduced to 100 m to shorten the simulation window to 6 ns, and TOD of the fibre is considered. The time delay of the interferometer is fixed at 1 ps thereby corresponding to the shear frequency of ~ 59.12 GHz. In the dataset, 54000 pulses are used for training, and the rest 6000 pulses are used for testing. We train the ISFC under the sampling rate of 100 GSa/s and the shorter fs pulse reconstruction results are shown in Fig. S5. The average duration MAE and the average phase RMSE over 6000 pulses in the test set are 0.01 fs and 0.071 rad respectively, manifest rather good pulse reconstruction.

Fig. S5 | Characterization over shorter fs pulses with LSSI-based ISFC. a. Pulse duration retrieval results of ISFC over the dataset with TOD, where the dashed line and shadow area respectively denote the ideal situation and the 10% error range. **b.** Temporal retrieval results (the left column) of ISFC over an 8-fs pulse and a 20-fs pulse from the test set, and corresponding spectral retrieval results (the right column) via performing Fourier transform on the retrieved temporal pulses.

Response 2.2:

We thank the reviewer for raising the question. The minimal pulse duration which can be measured by the LSSI-based method is independent of the back-end pulse reconstruction algorithm. The minimal pulse duration is determined by the bandwidth of the dispersive medium in the LSSI. We have revised the manuscript as follows.

Revision 2.2:**The third paragraph of Discussion**

The implementation of LSSI is flexible enough to support its utilization in various wavelengths. For instance, the spatial Michelson interferometer and diffraction gratings can also constitute the LSSI. Although dispersion is easier to obtain than nonlinearities for shorter wavelengths, the challenge of measuring ultraviolet fs pulses and even attosecond pulses via LSSI-based ISFC still lies in dispersion. LSSI requires the dispersive component providing large group velocity dispersion over a wide spectral range, covering the spectra of the pulses under measurement. Finding or designing the suitable dispersive component becomes a challenge. Certainly, the required dispersion amount can be reduced as the sampling rate increases. The LSSI-based ISFC can be applied to measure even shorter fs pulses accurately as shown in Fig. S5. The shortest pulse, which can be characterized by ISFC, is confined by the bandwidth of the dispersive medium in the LSSI. Therefore, given an ultra-large-bandwidth dispersive medium, single-shot full-field characterization over attosecond pulses via the LSSI-based ISFC can be expected.

Response 2.3:

We thank the reviewer for proposing the question. First, the group velocity dispersion has to satisfy the far-field Fraunhofer condition described by the inequation below for realizing DFT. t_{FWHM} is the temporal width. β_2 and L are the group velocity dispersion and length of the dispersive medium. The inequation sets a lower bound for the dispersion amount. It obvious that the wider pulses require larger group velocity dispersion to satisfy the far-field Fraunhofer condition. In the experimental dataset, the largest pulse duration is 3.21 ps. Thus, the smallest dispersion amount satisfying the far-field Fraunhofer condition is calculated to be $\sim 164 \text{ ps}^2$ (under the condition of $\frac{t_{FWHM}^2}{2\pi\beta_2L} = 0.01$), which corresponds to the dispersion of $\sim 290 \text{ ps/nm}$. We use a spool of DCF with the claimed group velocity dispersion of 340 ps/nm , closing to the low bound.

$$\frac{t_{FWHM}^2}{2\pi\beta_2L} \ll 1$$

On the other hand, when time delay is fixed, the dispersion determines the shear frequency as follows. Thus, when dispersion is too small, the shear frequency is very high and the requirements of the acquisition system (i.e., the photodetector and the oscilloscope) are very hard to be met. In the experiments, the claimed bandwidth of the photodetector (EOT ET-3500F) is $>15 \text{ GHz}$, and a real-time oscilloscope (Tektronix DPO70000SX Series) is used for interferogram acquisition at the sampling rate of 100 GSa/s and the bandwidth of 33 GHz . The current shear frequency is $\sim 2.13 \text{ GHz}$, which is much smaller than the bandwidth of the acquisition system and the sampling rate is enough. Further reducing the dispersion to the low bound of $\sim 290 \text{ ps/nm}$, the shear frequency rises up to $\sim 2.5 \text{ GHz}$ and the corresponding temporal interferograms can still be well acquired by the acquisition system. Therefore,

in our experiments, due to the high-speed acquisition system, the minimal dispersion amount is limited by the far-field Fraunhofer condition, which is ~ 290 ps/nm in our case.

$$\Delta\omega = \frac{\Delta t}{\beta_2 L}$$

The high-order dispersion effects do become increasingly significant when the pulse becomes even shorter. In the numerical simulations of characterization over shorter fs pulses with LSSI-based ISFC, TOD of the fibre is also considered and ISFC can still retrieve the pulse accurately as shown in Fig. S5.

Revision 2.3:

Experimental details and ISFC training in Methods.

The claimed group velocity dispersion of the DCF is -340 ps/nm. The dispersion amount has to satisfy the far-field Fraunhofer condition described by the inequation below for realizing DFT, where t_{FWHM} is the temporal width. β_2 and L are the group velocity dispersion and length of the dispersive medium. It sets a lower bound for the dispersion amount and the wider pulses require larger group velocity dispersion to satisfy far-field Fraunhofer condition. On the other hand, when time delay is fixed, the dispersion determines the shear frequency and too small dispersion causes the very high shear frequency. As a result, the demands on the sequential acquisition system become quite high and are very hard to be met.

$$\frac{t_{FWHM}^2}{2\pi\beta_2 L} \ll 1 \quad (10)$$

Comment 3:

3. Training of the AI Model

3.1 The authors briefly describe the training procedure in the Methods section. However, providing more detailed information about the model (such as the optimizer used, the number of layers, the number of epochs, and other relevant hyperparameters) would greatly enhance the readers' understanding.

3.2 The choice of a fully connected architecture is mentioned but not justified. Could the authors explain why this architecture was selected and what advantages it offers compared to alternatives (e.g., CNNs, RNNs, or hybrid models)?

3.3 The value of $\Delta\omega$ is expected to influence both the experimental results and the training performance of the AI model. Was this parameter fixed during the training process? If so, please clarify how it was chosen.

3.4 More detailed information about the preparation of training data would also improve the clarity of the manuscript. Could the authors describe in greater detail the data generation and preprocessing steps used to train the AI model?

Response 3.1:

We thank the reviewer for the advice. We have augmented the training part in Methods by providing more details of the model and training process.

Revision 3.1:

The third paragraph of Experimental details and ISFC training in Methods

The activation function is ReLU, and dropout is applied to alleviate overfitting. The fully-connected neural network is implemented using PyTorch, and key hyperparameters (e.g., the learning rate of 2.85e-4, training epochs of 500, the dropout ratio of 4.089e-2, and the batch size of 128) are optimized by Optuna. The optimizer is Adam and the cosine annealing learning rate strategy is applied for better convergence.

Response 3.2:

We thank the reviewer for raising the question. Fully-connected neural network provides the densest connection between two layers. The information of the previous layer can be transferred to the next layer without loss. The information transfer between two layers is totally determined by the learnable weights and bias, not the model structure. Therefore, we choose fully-connected neural network to implement ISFC and we have not tried other kinds of models.

Response 3.3:

We thank the reviewer for raising the question. The spectral shear $\Delta\omega$ is an imperative parameter in LSSI, which is determined by the time delay of the interferometer and the group velocity dispersion. In LSSI, the spectral shear can be expressed as

$$\Delta\omega = \frac{\Delta t}{\beta_2 L}.$$

Δt is the delay, β_2 and L are the group velocity dispersion and length of the dispersion medium respectively.

In most simulations, since the delay and dispersion are fixed, $\Delta\omega$ is fixed. Fixed spectral shear is friendly to the pulse reconstruction, especially when using FTM. However, delay jitter is inevitable in reality, resulting in varying $\Delta\omega$. In order to compare robustness of different pulse reconstruction algorithms against delay jitter, both FTM and ISFC are tested on a simulation dataset, where different levels of delay jitter are introduced, and the simulation result is shown in Fig. 3a. In this simulation, due to the delay is not fixed, $\Delta\omega$ is also not fixed. In experiments, delay jitter is induced by thermal and mechanical disturbances, also leading to a varying spectral shear.

The shear value is crucial to pulse reconstruction and the subsequent acquisition device. First, since the temporal fringe frequency is the shear when high-order dispersions are neglected, the temporal interferogram with the smaller shear can be sampled with a lower-bandwidth and lower-sampling-rate analog-to-digital converter (ADC). Second, given a fixed dispersive medium, the too small shear value corresponds to the too small delay, which is not enough to separate the pulse under measurement and its time-delayed replica in the time domain. As a result, there is overlapping hindering lossless sideband extraction after inverse Fourier transform in FTM, as shown in Fig. 3c. Hence, the optimal shear should be the minimal one that allows accurate pulse reconstruction. Overall, the choice of the shear value should consider both the bandwidth and sampling rate of the acquisition system and the approximate pulse duration of the pulse under measurement. In our experiments, the time delay is ~ 5.8 ps corresponding to the shear value of ~ 2.13 GHz. Considering the acquisition system in experiment has

the bandwidth of >15 GHz and the sampling rate of 100 GSa/s, the temporal fringes can be restored very well. But the time delay of ~5.8 ps is a little bit small for the widest pulse with the duration of 3.21 ps and it may cause overlapping. Fortunately, ISFC performs better under small time delays than FTM, which is manifested in simulation results shown in Fig. 3b, d, e. We have revised the manuscript accordingly for better understanding.

Revision 3.3:

The second paragraph of Numerical simulations of ISFC over fs pulses via LSSI

Because there is little energy outside the defined effective area, the corresponding phase is considered trivial to the pulse. Due to thermal and mechanical disturbances, fluctuations of key parameters in an LSSI system are inevitable in the real scene.

The third paragraph of Numerical simulations of ISFC over fs pulses via LSSI

Since the temporal fringe frequency is the shear when high-order dispersions are neglected, the temporal interferogram with the smaller shear can be sampled with a lower-bandwidth and lower-sampling-rate analog-to-digital converter (ADC). Given a fixed dispersive medium, the small shear value corresponds to the small delay, which may be not enough to separate the pulse under measurement and its delayed replica in the time domain. Therefore, the choice of the shear value should consider both the bandwidth and sampling rate of the acquisition system and the approximate pulse duration of the pulse under measurement. The optimal shear should be the minimal one that allows accurate pulse reconstruction.

Response 3.4:

We thank the reviewer for the advice. We have appended more detailed information about numerical simulations in the methods section as follows.

Revision 3.4:

Numerical simulation of LSSI in Methods

In numerical simulations, the temporal resolution is 10 fs and the time window is 20 ns. To generate pulses with asymmetric profiles, the fs pulse under characterization is synthesised by two randomly chirped Gaussian pulses, which can be expressed as

$$E_1(t) = A_1(t)e^{-\left(\frac{1+jC_{11}}{2}\left(\frac{t}{T_{01}}\right)^2 + \frac{jC_{12}}{6}\left(\frac{t}{T_{01}}\right)^3\right)}, \quad (6)$$

$$E_2(t) = A_2(t)e^{-\left(\frac{1+jC_{21}}{2}\left(\frac{t}{T_{02}}\right)^2 + \frac{jC_{22}}{6}\left(\frac{t}{T_{02}}\right)^3\right)}, \quad (7)$$

$$E(t) = E_1(t) + \alpha \cdot E_2(t - \tau). \quad (8)$$

C_{11} , C_{12} , C_{21} , C_{22} are random chirp parameter of two sub-pulses, T_{01} and T_{02} are random pulse durations of two sub-pulses. A random small delay τ is introduced between the two sub-pulses to create the asymmetric profile and α is the random attenuation factor. As a result, $E_1(t)$ serve as the main part and $E_2(t - \tau)$ is the secondary part of the synthesised fs pulse.

The fs pulse under characterization is centred at 1030 nm and first goes through the interferometer. Datasets with different delays ranging from 1 ps to 5 ps are generated. During the generation of the data with delay jitter, the delay is jittered randomly within the allowed maximal jitter magnitude for each pulse. By combining the raw pulse and its delayed replica, the spectral fringe patterns can be obtained.

Then, DFT is utilized to introduce a spectral shear between the two pulses and map the spectral fringes into time simultaneously. DFT is performed by a 3-km SM980 fibre with the group velocity dispersion of about -47.8 ps/nm/km at 1030 nm. We ignore nonlinearities in the fibre for faster data generation, and the dispersion can be modelled by a spectral phase filter as follows

$$E_{DFT}(\omega) = \mathcal{F}(E(t) + E(t - \Delta t)) \cdot e^{j\left(\frac{\beta_2 L}{2} \omega^2 - \frac{\beta_3 L}{6} \omega^3\right)}. \quad (9)$$

β_2, β_3 are the group velocity dispersion and TOD of the fibre respectively and L is the fibre length. $E_{DFT}(\omega)$ is the spectral field after DFT.

Comment 4:

4. Comparison between FTM and ISFC

4.1 Figures 2 and 5 present a comparison between the FTM and ISFC methods. The results indicate that ISFC is highly effective for LSSI-based pulse reconstruction, whereas the FTM method appears to perform very poorly. This raises concerns, as it is difficult to accept that FTM-based LSSI could be useful at all prior to the introduction of the AI-based approach. Could the authors elaborate on this point? Does the FTM method inherently suffer from such low accuracy, or might this comparison exaggerate its limitations? If the former is true, is it then an appropriate baseline for comparison?

4.2 In Figure 5, the authors compare ISFC with FROG. Why was a more detailed statistical analysis of the FROG results not performed? Such the analysis helps the robustness and generalizability of the proposed method.

Response 4.1:

We thank the reviewer for raising the question. FTM is indeed the standard demodulation algorithm in spectral shearing interferometry [Iaconis C, Walmsley I A. Spectral phase interferometry for direct electric-field reconstruction of ultrashort optical pulses[J]. *Optics Letters*, 1998, 23(10): 792-794]. Therefore, using FTM as the baseline is totally reasonable. Actually, FTM performs OK on the simulation datasets. Figure R3 shows two pulses well retrieved by FTM in the simulation dataset where the TOD is considered. The two pulses correspond two pink triangles in Fig. 2 in the manuscript.

Figure R3. Pulses retrieved by FTM in the simulation dataset where the TOD is considered.

When the TOD is absent in data generation, FTM performs even better as two pulses retrieval shown in Fig. R4 and the two pulses actually correspond two pink triangles in the top of Fig. 3d (Delay 5 ps) in the manuscript.

Figure R4. Pulses retrieved by FTM in the simulation dataset where the TOD is not considered.

Unfortunately, FTM performs really bad in experiments as shown in Fig. 5a. The most responsible reason for the terrible performance of FTM is its inherent sensitivity to the shear calibration, which has been unveiled in simulation results shown in Fig. 3a. Experiments are carried out in an open environment. Thus, during the long time of experimental data acquisition time due to the slow updating rate of the FROG (i.e., updating every ~ 7 seconds), the delay and dispersion parameters in LSSI determining the shear might vary along the environmental disturbances (e.g., thermal fluctuations and mechanical vibrations). However, the shear value used in FTM for reconstructing diverse pulses is unchanged, leading to awful pulse reconstruction. Another reason is that the shear frequency is too small. As validated in the simulation results shown in Fig. 3b,d, the small shear prevents effective sideband extraction as illustrated in Fig. 3c, thus leading to terrible pulse reconstruction of FTM in experiments.

Additionally, as we analyzed in the Discussion, non-negligible nonlinearities in fibre links of LSSI also contribute to the awful pulse reconstruction.

Response 4.2:

We thank the reviewer for the advice. We add statistics comparison to Fig. 5c.

Revision 4.2:

The second paragraph of Experimental validation of ISFC over fs pulses via LSSI.

Figure 5a shows the pulse duration retrieval accuracy comparison over the test set. The superiority of ISFC over FTM is further magnified in experiments, manifesting decent pulse duration retrieval from 146 fs to 3.21 ps. The duration MAE on the experimental test set of ISFC is 41.32 fs, which is even smaller than the pre-set temporal resolution of ~ 70 fs of the FROG in experiments. However, the duration MAE on the experimental test set of FTM reaches 8.16 ps. As shown in Fig. 5c, the corresponding duration NMAE of ISFC and FTM are ~ 0.07 and 17.12 respectively. In terms of the phase retrieval accuracy, the phase NRMSE of ISFC is 0.0256, which is ~ 17 folds better than that of FTM.

Figure 5

Fig. 5 | Comparison of FTM and ISFC for fs pulse reconstruction in experiments. **a.** Pulse duration retrieval results of FTM and ISFC, where the dashed line and shadow area respectively denote the ideal situation and the 10% error range. **b.** Accurate pulse reconstruction of ISFC generalizes to the short pulse (top row), pulses with different signs of chirp (middle rows), and the pulse with a sophisticated shape (bottom row). **c.** Statistics comparison between FTM and ISFC. The duration NMAE of ISFC and FTM are ~ 0.07 and 17.12 respectively. The phase NRMSE of ISFC and FTM are 0.0256 and 0.4337 respectively.

Comment 5:

5. Switching Dynamics

5.1 Figure 6 presents the switching dynamics of the programmable spectral filter. The results show excellent agreement between ISFC and FROG, which strongly supports the validity of the characterization. However, it would be helpful if the authors could clarify in which specific applications or scenarios such switching characteristics could be practically utilized.

5.2 What is the shortest pulse duration for which this switching dynamic can be reliably demonstrated? Please provide a discussion of the limits.

Response 5.1:

We thank the reviewer for the advice. First, the proposed method can help to simplify the systems in spectroscopy-related applications. For instance, fs pulses can be used to measure the spectral response of various matters (e.g., gases). For absorption response measurement, we only need to measure the spectra of the fs pulses before and after going through the matters under test. To obtain the phase response is more complex, coherent detection is required. The fs pulses are split into two branches with one branch goes through the matters under test and the other serves as the reference. Thus, the phase difference between the two branches, which is also the phase response of the matters under test, can be demodulated. With the proposed LSSI-based ISFC, the phase of fs pulse itself can be accurately retrieved. Therefore, fs pulses can directly go through the matters under test and do not need to introduce interferometry during testing. The phase of fs pulses before and after going through the matters under test can be retrieved by the proposed method and the phase response under test lies in the change of the fs pulses' phase. On the other hand, the measure speed can be substantially enhanced by the LSSI-based ISFC. Conventional ultrafast spectroscopy relies on spectrometers for spectra readout, which is quite slow thereby leading to slow measure speed. However, the LSSI-based ISFC retrieve fs pulses from temporal interferograms. As a result, the measure speed is be substantially enhanced and can easily reach Mega frames per second, making the LSSI-based ISFC suitable for recording high-speed events.

Second, ultrafast laser dynamics studies enabled by DFT is very popular [1-8]. However, these studies mainly focus on spectral intensity evolution and the phase evolution is unavailable [1-6]. Time lens and dispersive Fourier transform (DFT) are utilized together for full-field characterization over picosecond pulses by performing the iterative Gerchberg-Saxton algorithm between temporally magnified pulse profiles and real-time spectra. Thus, the full-field characterization of transient dissipative soliton dynamics and intracavity incoherent supercontinuum dynamics are realized [7,8]. With the proposed LSSI-based ISFC, the phase evolution is readily available and full-field characterization of ultrafast laser dynamics can be achieved with a linear, simple physical setup.

[1] Runge A F J, Broderick N G R, Erkintalo M. Observation of soliton explosions in a passively mode-locked fiber laser[J]. *Optica*, 2015, 2(1): 36-39.

[2] Herink G, Jalali B, Ropers C, et al. Resolving the build-up of femtosecond mode-locking with single-shot spectroscopy at 90 MHz frame rate[J]. *Nature Photonics*, 2016, 10(5): 321-326.

[3] Herink G, Kurtz F, Jalali B, et al. Real-time spectral interferometry probes the internal dynamics of femtosecond soliton molecules[J]. *Science*, 2017, 356(6333): 50-54.

[4] Liu X, Yao X, Cui Y. Real-time observation of the buildup of soliton molecules[J]. *Physical Review Letters*, 2018, 121(2): 023905.

[5] Liu X, Pang M. Revealing the buildup dynamics of harmonic mode-locking states in ultrafast lasers[J]. *Laser & Photonics Reviews*, 2019, 13(9): 1800333.

[6] Liu X, Popa D, Akhmediev N. Revealing the transition dynamics from Q switching to mode locking in a soliton laser[J]. *Physical Review Letters*, 2019, 123(9): 093901.

[7] Ryzkowski P, Närhi M, Billet C, et al. Real-time full-field characterization of transient dissipative soliton dynamics in a mode-locked laser[J]. *Nature Photonics*, 2018, 12(4): 221-227.

[8] Meng F, Lapre C, Billet C, et al. Intracavity incoherent supercontinuum dynamics and rogue waves in a broadband dissipative soliton laser[J]. *Nature Communications*, 2021, 12(1): 5567.

Revision 5.1:

The fourth paragraph of Experimental validation of ISFC over fs pulses via LSSI.

ISFC provides a viable solution to capture ultrafast events in a single-shot and full-field manner at a frame rate up to MHz. As a result, the LSSI-based ISFC can find utilizations in diverse scenes, such as ultrafast spectroscopy²⁵ and full-field characterization of ultrafast laser dynamics^{12,13,26-28}.

References

25. Maiuri M, Garavelli M, Cerullo G. Ultrafast spectroscopy: State of the art and open challenges[J]. *Journal of the American Chemical Society*, 2019, 142(1): 3-15.

26. Herink G, Jalali B, Ropers C, et al. Resolving the build-up of femtosecond mode-locking with single-shot spectroscopy at 90 MHz frame rate[J]. *Nature Photonics*, 2016, 10(5): 321-326.

27. Herink G, Kurtz F, Jalali B, et al. Real-time spectral interferometry probes the internal dynamics of femtosecond soliton molecules[J]. *Science*, 2017, 356(6333): 50-54.

28. Liu X, Pang M. Revealing the buildup dynamics of harmonic mode-locking states in ultrafast lasers[J]. *Laser & Photonics Reviews*, 2019, 13(9): 1800333.

Response 5.2:

We thank the reviewer for raising the question. The switching dynamics is resolved by the LSSI-based ISFC frame by frame. Concretely, a piece of continuous temporal waveform containing the switching is firstly segmented into numerous frames. Each frame contains a temporal interferogram, which is generated by one fs pulse and its delayed replica during switching. Then, the segmented frames are sent to ISFC to retrieve each fs pulse during switching. Therefore, the shortest pulse duration in switching dynamics is determined by the ability of the proposed LSSI-based ISFC. As explained previously, the minimal pulse duration which can be measured by the LSSI-based method is independent of the back-end pulse reconstruction algorithm. The minimal pulse duration is determined by the bandwidth of the dispersive medium in the LSSI.

Comments from Reviewer #3:

In the manuscript, the authors demonstrate the intelligent single-shot full-field characterization of femtosecond pulses based on linear spectral shearing interferometry. Since it is a linear process, the work significantly reduced laser pulse energy compared with methods that employ nonlinear process for laser pulse measurements such as SPIDER and FROG. By employing machine learning, the work obviously reduced the affect by noise. The manuscript is well written and the results are interesting. Before I recommend the manuscript accept by Nature Communications, I would like to ask the authors to consider my concerns below for further improvement of the manuscript.

Comment 1:

In the manuscript, the signal is detected employing photo detector and oscilloscope. But is the sampling speed enough since the measured pulse is femtosecond and the dispersed pulse duration is in ps.

Response 1:

We sincerely thank the reviewer for the general comments and proposing the questions. The dispersed signal is in nanosecond level as shown in the figure below, which can be readily acquired by high-speed photodetector and oscilloscope.

Figure R5. **Top**, Spectral interferogram before DFT; **Bottom**, 161 segmented temporal frames (grey) and their coherent average result (black).

Revision 1:**The second paragraph of Experimental details and ISFC training in Methods.**

The dispersed signal reaches nanosecond-level width in time domain (see Fig. S3 in Supplementary Information) and it can be readily acquired by the high-speed real-time acquisition system. The bandwidth of the photodetector (EOT ET-3500F) is >15 GHz.

Comment 2:

What is the pulse measurement accuracy and how short the pulse can be measured? For even shorter pulses, GDD, TOD and FOD are all important for the pulse measurement, but the work only measured up to TOD.

Response 2:

We thank the reviewer for proposing the question. We add statistics comparison to Fig. 5c. In the experimental test dataset with the pulse durations ranging from 146 fs to 3.21 ps, the duration MAEs and the phase NRMSE of ISFC is 41.32 fs and 0.0256 respectively, signifying rather decent pulse reconstruction. The minimal pulse duration which can be measured by the LSSI-based method is independent of the back-end pulse reconstruction algorithm. The minimal pulse duration is determined by the bandwidth of the dispersive medium in the LSSI.

Indeed, high-order dispersion terms are important for even short fs pulses. However, when the experimental setup is fixed, all dispersion terms can be considered nearly unchanged. Unlike FTM, the proposed ISFC does not need dispersion calibration and it can sense all dispersion terms through training with numerous data. As in our experiments, the dispersion medium is a spool of dispersive compensation fibre and it also contains high-order dispersion terms (i.e., TOD, FOD). But ISFC can still accurately reconstruct fs pulses from temporal interferograms without knowing the exact high-order dispersion values.

Comment 3:

In the manuscript, the authors claimed that the method can measure attosecond pulses. Is it possible since the measurement accuracy requires is quite high. How reliable is it? Is there any suitable dispersion materials can provide sufficient dispersion?

Response 3:

We thank the reviewer for proposing the question. The proposed method can measure attosecond pulses in principle. However, there are two major challenges in achieving it. First, as we mentioned in the manuscript, it requires an ultra-large-bandwidth dispersive medium and we now do not know what kind of material can provide broadband dispersion for attosecond pulse characterization. Second, ISFC is trained in supervising manner currently. As a result, it requires labelled dataset for ISFC training and the measurement accuracy is highly related to the accuracy of the labelled dataset.

Comment 4:

What is the requirement on training data accuracy, which is quite important for characterizing ultrashort laser pulses with high reliability?

Response 4:

We thank the reviewer for proposing the question. We suppose the training data obtained by a well-calibrated fs pulses measurement system has decent measurement accuracy and is acceptable. The training data of ISFC is generated with a commercial FROG (MesaPhotonics FROGScan), whose parameters are shown below. The temporal resolution is below 2 fs and it is suitable for the experimental dataset ranging from 146 fs to 3.21 ps. During data collection in experiments, the FROG trace errors (i.e., the error between the measured FROG trace and the retrieved FROG trace) of most pulses is less than 2%, which signifying reliable measurements.

Parameter	Specification
Input Pulse Wavelength Range	450 nm - >2000 nm
Pulse Length Range	< 15 fs - 12 ps
Temporal Range	30 ps
Temporal Resolution	2 fs or better
Delay Increment	1 fs
Spectral Resolution	0.20 nm - 1 nm
Spectral Range	100 nm - 600 nm
Pulse Complexity	TBWP > 50
Intensity Accuracy	2%
Phase Accuracy	0.01 radians
Real-time Sensitivity (IpeakIave)	4 W ²
Averaged Sensitivity (IpeakIave)	< 0.1 W ²
Input Beam Size	2 - 8mm collimated
Nominal Polarization	Horizontal (vertical by rotating crystal)
Acquisition Speed	> 1 Hz 64 x 64 grid
Spectra required for measurement	Number in grid

Figure R6. FROG parameters.

Comment 5:

How much amount dispersion is required for the measurement, considering the temporal resolution of detector and oscilloscope.

Response 5:

We thank the reviewer for proposing the question. First, the dispersion has to satisfy the far-field Fraunhofer condition described by the inequation below for realizing DFT. t_{FWHM} is the temporal width. β_2 and L are the group velocity dispersion and length of the dispersive medium. The inequation sets a lower bound for the dispersion amount. It obvious that the wider pulses require larger group velocity dispersion to satisfy the far-field Fraunhofer condition. In the experimental dataset, the largest pulse duration is 3.21 ps. Thus, the smallest dispersion amount satisfying the far-field Fraunhofer condition is calculated to be $\sim 164 \text{ ps}^2$ (under the condition of $\frac{t_{FWHM}^2}{2\pi\beta_2L} = 0.01$), which corresponds to the dispersion of $\sim 290 \text{ ps/nm}$. We use a spool of DCF with the claimed group velocity dispersion of 340 ps/nm, closing to the low bound.

$$\frac{t_{FWHM}^2}{2\pi\beta_2L} \ll 1$$

On the other hand, when time delay is fixed, the dispersion determines the shear frequency as follows. Thus, when dispersion is too small, the shear frequency is very high and the requirements of the acquisition system (i.e., the photodetector and the oscilloscope) are very hard to be met. In the experiments, the claimed bandwidth of the photodetector (EOT ET-3500F) is $>15 \text{ GHz}$, and a real-time oscilloscope (Tektronix DPO70000SX Series) is used for interferogram acquisition at the sampling rate of 100 GSa/s and the bandwidth of 33 GHz. The current shear frequency is $\sim 2.13 \text{ GHz}$, which is much smaller than the bandwidth of the acquisition system and the sampling rate is enough. Further reducing the dispersion to the low bound of $\sim 290 \text{ ps/nm}$, the shear frequency rises up to $\sim 2.5 \text{ GHz}$ and the corresponding temporal interferograms can still be well acquired by the acquisition system. Therefore, in our experiments, due to the high-speed acquisition system, the minimal dispersion amount is limited by the far-field Fraunhofer condition, which is $\sim 290 \text{ ps/nm}$ in our case.

$$\Delta\omega = \frac{\Delta t}{\beta_2 L}$$

Revision 5:**Experimental details and ISFC training in Methods.**

The claimed group velocity dispersion of the DCF is -340 ps/nm. The dispersion amount has to satisfy the far-field Fraunhofer condition described by the inequation below for realizing DFT, where t_{FWHM} is the temporal width. β_2 and L are the group velocity dispersion and length of the dispersive medium. It sets a lower bound for the dispersion amount and the wider pulses require larger group velocity dispersion to satisfy far-field Fraunhofer condition. On the other hand, when time delay is fixed, the dispersion determines the shear frequency and too small dispersion causes the very high shear frequency. As a result, the demands on the sequential acquisition system become quite high and are very hard to be met.

$$\frac{t_{FWHM}^2}{2\pi\beta_2 L} \ll 1 \quad (10)$$

Comment 6:

In the work, sampling rate is crucial thus high repetition rate is required. What is the lowest repetition rate that can be accepted. What is the limitations of the method.

Response 6:

We thank the reviewer for proposing the question. The proposed ISFC realizes single-shot full field characterization over fs pulse. Thus, there is no low bound for the repetition rate of fs pulses. However, to guarantee good pulse reconstruction, the stretched interferograms cannot overlap after DFT. Therefore, the upper bound of the repetition rate is limited by the dispersion.

As for the sampling rate, an empirical deduction drawn from the simulation (see Supplementary Information) and experimental results indicates that the minimal sampling rate supported by ISFC lies between the actual shear value and its corresponding Nyquist sampling rate. Though we do not suppose ISFC violates the Nyquist sampling theorem. Because the shear value remains unchanged in simulations and it can be learnt by ISFC through numerous data. When the sampling rate is below the Nyquist sampling rate, ISFC automatically pads the missing but deterministic data points during computation. The shear value in experiments varies due to the environmental disturbances. However, the variation is rather small as the fringe frequencies of different temporal interferograms are all very close to the calibrated shear value of ~2.13 GHz. As a result, the pulse reconstruction performance of ISFC is acceptable under the sampling rate of 3.13 GSa/s. On the other hand, the performance drop induced by using the sampling rate below the Nyquist sampling rate is quite obvious in both simulations and experiments.

Revision 6:**The fifth paragraph of Experimental validation of ISFC over fs pulses via LSSI**

The pulse retrieval performance sharply deteriorates when further reducing the sampling rate to 1.56 GSa/s, as evidenced by the duration NMAE drastically increasing by 104.1% compared to the sampling rate of 100 GSa/s. An empirical deduction drawn from the simulation and experimental results indicates

that the minimal sampling rate supported by ISFC lies between the actual shear value and its corresponding Nyquist sampling rate (see Supplementary Information).

Comment 7:

Can the method measure electric waveform of the laser pulses, which really requires high accuracy and resolution, and there are works have reported such as TIPTOE, FROG and SPIDER.

Response 7:

We thank the reviewer for proposing the question. The electric field of fs pulses can be expressed as follows

$$E_e(t) = \sqrt{I(t)}e^{j(\omega_0 t + \phi_{CEP} + \phi_{env}(t))}.$$

ω_0 is the central angular frequency. $I(t)$ and $\phi_{env}(t)$ are the intensity envelope and envelope phase of fs pulse, which can be measured by FROG, SPIDER or the LSSI-based ISFC. ϕ_{CEP} is the carrier envelope phase, which determines the temporal relation of the pulse envelope with respect to the underlying carrier oscillation. Most pulse characterization methods (e.g., FROG, SPIDER) are not able to measure the absolute value of ϕ_{CEP} . The absolute value of ϕ_{CEP} is not important if the pulse envelope does not significantly vary within one oscillation period. We set $\phi_{CEP} = 0$ and the electric field can be retrieved as shown in Fig. R7, where the top plots show the measured intensity envelope and envelope phase of a fs pulse by FROG and the LSSI-based ISFC and the bottom shows the retrieved electric field.

Figure R7. The retrieved electric field of a fs pulse.

We sincerely thank the reviewers for their careful reviewing of the manuscript entitled “Intelligent single-shot full-field characterization over femtosecond pulses” by Guoqing Pu, Chao Luo, Weisheng Hu, and Lilin Yi. We appreciate their invaluable comments. We have revised the manuscript according to the comments of the reviewers. The revised manuscript is provided. For your convenience, we provide this response letter with the reviewer’s comments in *italic*, with our revisions underlined and **highlighted in yellow**. In the following, we answer the comments raised by the reviewer in detail.

Comments from Reviewer #1:

I am satisfied with the revised version and can recommend acceptance.

Response:

We sincerely thank the reviewer for raising previous questions, which really helps us to improve the quality of the manuscript, and we thank the reviewer feels satisfied with our revision.

Comments from Reviewer #2:

The authors have provided a comprehensive and well-prepared rebuttal that addresses the majority of the reviewers’ comments in a clear and scientifically rigorous manner. The revised manuscript has been substantially improved both in clarity and completeness.

The proposed LSSI-based intelligent single-shot full-field characterization (ISFC) represents a technically innovative approach that combines linear spectral shearing interferometry with data-driven reconstruction. The idea of employing a neural network to circumvent calibration instabilities and high-order dispersion issues is novel and potentially impactful.

However, I still have one concern regarding short (~10 fs level) pulse laser with conventional dispersive medium that causing group delay dispersion. (Practical point of view and precision.)

The authors claim that ISFC can characterize pulses as short as 8 fs based on simulations including TOD. However, in practice, broadband femtosecond pulses inevitably experience strong dispersion and spectral phase distortion through the DCF or grating used in LSSI. Such high-order dispersion leads to non-linear temporal-to-spectral mapping that may not be easily compensated by data-driven approaches trained on limited ranges of ω and k . Therefore, it remains unclear whether the proposed technique can practically characterize sub-20 fs pulses under realistic experimental conditions. The authors are encouraged to discuss this limitation more concretely or provide preliminary experimental verification with broader-bandwidth pulses.

Response:

We sincerely thank the reviewer for the positive comments on our previous revisions and presenting the professional concerns about the high-order dispersions when the pulse is too short. As the reviewer mentions, broadband femtosecond pulses inevitably experience spectral phase distortion in strong dispersion, where high-order dispersions lead to nonlinear temporal-to-spectral mapping. However, as shown in Equation S1, the high-order dispersions induced nonlinear temporal-to-spectral mapping can be precisely expressed as a polynomial combination between the temporal and spectral coordinates, with the coefficients determined by the dispersion parameters.

$$f = f_0 + \frac{1}{2\pi\beta_2 L} t - \sum_{k=3}^{\infty} \frac{\beta_k L}{2\pi(k-1)!(\beta_2 L)^k} t^{k-1}. \quad (S1)$$

In realistic experiments, though high-order dispersion parameters are not easy to be characterized accurately, learning such a polynomial combination through massive data using AI is not a hurdle. Due to the limitation on our laser source, we feel sorry that we cannot provide additional experimental results.

Revision:

The first paragraph of Numerical simulations of ISFC over fs pulses via LSSI.

When the pulse becomes even shorter, the conventional TOD compensating method cannot completely compensate for the TOD-induced nonlinear mapping effect. As shown in Equation (S1) in Supplementary Information, the high-order dispersions induced nonlinear spectral-to-temporal mapping can be expressed as a polynomial combination between the temporal and spectral coordinates, with the coefficients determined by the dispersion parameters. Thus, the nonlinear polynomial relation can be accurately learnt by ISFC through training on massive data, waiving precise calibration on the high-order dispersion parameters. As a result, ISFC performs perfectly in the entire pulse duration range.

Comments from Reviewer #3:

The authors as carefully respond to my comments and make revisions to the manuscript. I agree that the manuscript can be accepted now.

Response:

We sincerely thank the reviewer for raising previous questions, which really helps us to improve the quality of the manuscript, and we thank the reviewer feels satisfied with our revision.

Other revisions:

We have checked the whole manuscript again and we have corrected several typos as below.

Institution

State Key Laboratory of Photonics and Communications, School of Information Science and Electronic Engineering, Shanghai Jiao Tong University, Shanghai 200240, China

Equation (4)

$$\Gamma_{err} = \frac{1}{N} \cdot \sum_{n=1}^N \frac{|\tau_n - \widehat{\tau}_n|}{\widehat{\tau}_n}, \quad (4)$$

Equation (9)

$$E_{DFT}(\omega) = \mathcal{F}(E(t) + E(t - \Delta t)) \cdot e^{j\left(\frac{\beta_2 L}{2} \omega^2 + \frac{\beta_3 L}{6} \omega^3\right)}. \quad (9)$$